

# Volcanic Aerosol Modification of the Stratospheric Circulation in E3SMv2 Part I: Wave-Mean Flow Interaction

Joseph P. Hollowed[1], Christiane Jablonowski[1], Thomas Ehrmann[2], Diana Bull[2], Benjamin Wagman[2], and Benjamin Hillman[2]

[1]Department of Climate and Space Sciences and Engineering, University of Michigan, Ann Arbor, MI, USA
[2]Sandia National Laboratories, Albuquerque, NM, USA

**Correspondence:** Joseph Hollowed (hollowed@umich.edu)

**Abstract.** Following tropical volcanic eruptions, westerly zonal wind accelerations have been observed in the winter hemisphere polar vortex region. This wind response has been reproduced in some (but not all) simulated eruption studies. As the primary effect of volcanic aerosols is to heat the tropical stratosphere, the midlatitude zonal wind response is often explained as thermal wind effect. Several studies have shown that this explanation is insufficient in understanding the relative significance of the aerosol direct effect, and indirect dynamical feedbacks. In this work, we use a Transformed Eulerian Mean (TEM) framework to identify the dynamical origins of stratospheric wind anomalies following the simulated 1991 eruption of Mt. Pinatubo. A paired set of volcanic and non-volcanic 15-member ensembles is used to isolate the volcanic impact. A TEM decomposition of the net zonal wind forcing is then performed to close the differenced momentum budget between the two ensembles. Zonal wind accelerations near 30–40N and 3–30 hPa are identified with significance in the Northern Hemisphere (NH) during both the summer and winter. We find each of these seasonal acceleration episodes to have distinct dynamical drivers. In the summertime, the response is primarily governed by an accelerated meridional residual circulation. In the wintertime, the response is eddy-driven, where an equatorward deflection of planetary waves was robustly identified near 30N and 30 hPa. We additionally identified that a deficit of wave forcing in the tropical stratosphere dampens the amplitude of the quasi-biennial oscillation (QBO) for at least two years following the eruption.

## 1 Introduction

On seasonal to interannual timescales, the mean-flow of the stratosphere exhibits a remarkable diversity of states. The most important modes of stratospheric variability, such as the dramatic development and deterioration of the winter-time polar jets, the oscillation of tropical zonal winds, and the seasonal reversal of the mean-meridional circulation of mass, are highly consequential on the general circulation of the atmosphere (Butchart, 2022). These phenomena also exert influence on conditions near the surface by stratosphere-troposphere coupling, affecting tropospheric weather predictability (Boville and Baumhefner, 1990; Baldwin and Dunkerton, 2001; Scaife et al., 2022).

Much of observed stratospheric variability arises from an internal, dynamical origin. At the same time, there are significant drivers of variability due to sources exogenous to the Earth system, which act to alter the stratosphere's structure and chemical





composition. In addition to the anthropogenic depletion and recovery of ozone, one of the most significant natural sources of

external forcing on the stratosphere are volcanic eruptions (Schurer et al., 2013). Very large eruptions expel enormous quantities

of sulfur dioxide ($SO_2$) into the free atmosphere, which oxidize to form a long-living population of sulfate aerosols (Bekki,

1995). Once a volcanic aerosol plume is delivered to the upper stratosphere by its own self-lofting (Stenchikov et al., 2021)

and the tropical pipe (Kremser et al., 2016), it gradually becomes globally-distributed via the Brewer-Dobson Circulation

(BDC; Butchart (2014)). Detectable stratospheric aerosol concentrations will then persist for years. Meanwhile, absorption

of longwave radiation by sulfate aerosols drives an increase of tropical stratospheric temperatures, establishing an enhanced

equator-to-pole temperature gradient.

Volcanic aerosol-induced temperature perturbations of this nature must give rise to subsequent perturbations in wind—

and thus the zonal-mean stratospheric circulation as a whole—such that adiabatic cooling approximately balances diabatic

heating in the tropics, and midlatitude thermal-wind balance is maintained at low Rossby number. While the major modes

of stratospheric variability such as the quasi-biennial oscillation (QBO) and the arctic/antarctic oscillations (AO/AAO) and

their governing mechanisms are well-described in the literature, their quantitative details are controlled by highly nonlinear

combinations of forcing terms. Thus, from a theory standpoint, it is difficult to ever know *a priori* how they will respond to a

transient source of external forcing, such as radiative heating by volcanic aerosols.

Studies of volcanically-driven changes to the stratospheric circulation typically focus on either the average response to an

ensemble of eruptions, or to singular volcanic events in the historical record. Particular attention has been paid to the 1991

eruption of Mt. Pinatubo in the Philippines, which released 15–20 Tg of $SO_2$, and induced middle-stratosphere temperature

anomalies of a few Kelvin for several years following the event (Self et al., 1997; McCormick et al., 1995). Early observational

analyses by Kodera (1994) and Graf et al. (1994) identified a strengthening of the Northern Hemisphere (NH) polar vortex

during boreal winter of 1991-1992, despite presence of El-Ninõ conditions (otherwise associated with a weakened vortex).

This vortex-strengthening effect is correlated to an enhanced AO (see e.g. Baldwin and Dunkerton (1999)), which itself has

been observed by an empirical orthogonal function (EOF) analysis of sea-level pressure data following 13 major volcanic

eruptions between 1873 and 2000 (Christiansen, 2008).

Accordingly, positive polar vortex anomalies, and an excitement of the positive AO more generally, are often used as a

qualitative benchmark of a climate model's volcanic response. Barnes et al. (2016) used an average of 13 separate model simu-

lations contributed to the Fifth Coupled Model Intercomparison Project (CMIP5; Taylor et al. (2012)), and found a statistically

significant strengthening of the polar vortex in austral (boreal) winter of 1991 (1992), accompanied by poleward shifts of the

tropospheric jet stream in both hemispheres until February of 1992, suggesting an enhanced AO and AAO. This same result has

been found in some additional Pinatubo model studies (Stenchikov et al., 2002; Karpechko et al., 2010; Bittner et al., 2016),

but has also been weaker or missing from others (Stenchikov et al., 2006; Driscoll et al., 2012; Toohey et al., 2014; Polvani

et al., 2019).

The absence of a robust vortex enhancement is not necessarily indicative of a model's inability to capture the mechanisms

which govern that response in nature. Rather, previous studies (Stenchikov et al., 2002, 2006; Toohey et al., 2014) emphasize

that this apparent failure is often just a sampling problem; variability of the polar vortex (and thus the AO) is indeed a function of



the volcanic forcing structure, but also of internal variability and feedbacks. In the midlatitudes, these effects are comparable in magnitude, and so an ensemble of numerical experiments will not be guaranteed to match any single realization. The conclusion is that in some cases, either a large ensemble size or a very large volcanic eruption may be required for a significant response to be observed. This idea is supported by the robust vortex enhancements simulated in the 700 Tg experiments of Toohey et al. (2011). Likewise, Bittner et al. (2016) found a robust positive vortex response for a 55 Tg eruption, but a weak response and large inter-ensemble spread for a Pinatubo-like eruption with an otherwise comparable experimental setup.

In the tropics, there have also been comparisons between model responses and post-Pinatubo observations. Kinne et al. (1992) reported an increase in observed tropical upwelling, which modifies the vertical wind structure, and by mass continuity, is associated with an accelerated BDC. This effect was previously suggested as early as Dunkerton (1983), and has since been observed in simulated environments (DallaSanta et al., 2021; Brown et al., 2023), where the consequences of this upwelling on the QBO were studied.

Overall, while some consensus has developed around the qualitative circulation response to volcanic forcing, the causes for differences in the quantitative details between models often remain uncertain. From a process-level point of view, however, we may still be interested in asking: What are the dynamical mechanisms which give rise to a particular post-volcanic state, in a particular model?

An instinctive answer to this question is that the zonal wind must remain in balance with the aerosol-induced temperature perturbations, and thus the enhanced equator-to-pole temperature gradient drives accelerated winds from the subtropics to the midlatitudes. This is the well-known "thermal wind balance" hypothesis, which has been shown to be insufficient by several authors (Toohey et al., 2014; Bittner et al., 2016; DallaSanta et al., 2019). While it is true that the post-eruption atmosphere must indeed be in thermal wind balance at low Rossby number, it is also *not* the case that the net temperature response is solely aerosol-induced. Rather, the net temperature response is a result of aerosol-driven heating, and the nonlinear feedbacks that follow.

The idealized model hierarchy studies of DallaSanta et al. (2019) clearly demonstrated that a zonal-wind field artificially constrained to respond only to the aerosol-induced temperature adjustment is unable to produce accelerations of the NH vortex, shifts of the tropospheric jets, or other features associated with an enhanced AO. Among several factors tested, they show that three-dimensional eddy-driven feedbacks are crucial to establishing the expected zonal-wind response patterns, and thus the balance between temperature and an accelerated extratropical vortex region is only known *a posteriori*.

In an effort to clarify precisely how midlatitude eddies mediate the volcanic response, Bittner et al. (2016) ran a 20-member ensemble of eruption experiments, and analyzed the differences in planetary wave propagation with respect to volcanically-quiescent control runs. They found that the first-order response to the tropical temperature perturbations is an enhancement of westerlies not in the vortex region, but at lower latitudes, near 30°N and 10 hPa. This amounts to a change in the background condition for wave propagation, causing an anomalous equatorward deflection of planetary waves, and thus hampered wave dissipation in the vortex region aloft and poleward. The net result was increased vortex wind speeds during boreal winter following the volcanic event. This effect was observed with weak significance for a Pinatubo-sized eruption, but was shown decisively for an eruption of about 55 Tg $SO_2$.





In a similar experiment by Toohey et al. (2014) using a 16-member ensemble of Pinatubo simulations, it was likewise found
that enhanced wave drag occurs near 30°N and 10 hPa during boreal winter of 1991–1992, though this did not manifest in
a statistically-significant vortex enhancement. The authors further relate the modified large-scale wave activity to balanced
changes in the residual meridional circulation, which suggests an acceleration of the BDC.

Incidentally, this observation immediately implies the observed anomalous upwelling in the tropics, and thus coupling to
the QBO. In this region, there has also been recent progress made on understanding the driving mechanisms of the volcanic
response. In their simulations, Brown et al. (2023) observed that the vertical advection of momentum associated with enhanced
upwelling causes a delay of the descending QBO phase, effectively elongating the QBO period for several years following a
Pinatubo-like event. Due to feedbacks with the QBO secondary circulation, this delay more strongly affects the QBO in a state
of easterly shear than westerly, and the response is thus highly sensitive to the QBO state at the time of eruption.

The essential takeaway from this summary of the literature is that the fundamental atmospheric response to a tropical vol-
canic eruption is a modification of (predominantly surf-zone) wave activity, and the associated modification to the meridional
residual circulation. In this view, specific terms like "vortex strengthening" are perhaps imprecise. For example, volcanically-
enhanced midlatitude westerlies are sometimes shown to align with the vortex core, but are often instead shown to align with
the equatorward edge of the vortex. We suggest that a description of this response as being either a "vortex strengthening" or
a "vortex shift" is really a distinction without a difference, as far as the driving processes are concerned, and that we should
instead consider the momentum budget more generically. This idea is also supported by the fact that the "vortex region"
anomalies are often shown to be hemispherically symmetric, despite the fact that the southern hemisphere (SH) stratosphere is
quiescent during boreal winter (DallaSanta et al., 2019).

In this work, we examine the response to the simulated eruption of Mt. Pinatubo from a Transformed Eulerian Mean (TEM)
perspective, in a single coupled climate model. The primary mechanisms controlling large-scale transport of momentum in the
stratosphere are advection by the so-called residual (diabatic) circulation, as well as vertical and horizontal wave propagation
and dissipation. The TEM framework defines a zonal momentum budget, in which the net local forcing is understood as
a combination of these mechanisms. The present goal is to close the TEM budget within regions of interest, in order to
understand precisely the dynamical processes which control the development and deterioration of statistically significant post-
volcanic wind anomalies. Anomalies are defined as the difference between paired sets of runs, with and without the eruption
activated, and otherwise identical initial conditions. The aerosol treatment is prognostic, and so the divergence of each pair
of runs is due to the aerosol forcing, as well as feedbacks to the aerosol distribution. All ensemble members are seeded via
precision-level perturbations of a common initial state, which reflects the observed conditions of the real-world Pinatubo event.
In this way, we are explicitly ignoring confounding factors such as differences in major climate modes at the time of eruption
such as the El-Niño-Southern Oscillation (ENSO) and the QBO phases.

In Sect. 2, we describe the climate model employed, and the simulation ensembles. In Sect. 3, we define the recipes for our
statistical ensemble measures, and the TEM formalism. Section 4 shows the characteristics of the background (volcanically-
quiescent) runs, and Sect. 5 presents the results of the experiments. Section 6 and 7 provides a discussion of our results in
relation to previous works.





## 2 Simulations

The numerical experiments utilized for this study were conducted in a custom version of the Energy Exascale Earth System
Model version 2 (E3SMv2; Golaz et al. (2022)) called E3SMv2-SPA, described in Brown et al. (2024). While E3SMv2 de-
scribes volcanic eruptions by a prescribed forcing from the GloSSAC reanalysis dataset (Thomason et al., 2018), E3SMv2-SPA
instead replaces this treatment with a stratospheric prognostic aerosol (SPA) capability. Rather than prescribing stratospheric
light extinction directly, $SO_2$ is emitted as a tracer into the stratosphere, which forms a sulfate aerosol as governed by the
tuned, four-mode version of the Modal Aerosol Module (MAM4; Liu et al. (2016)). Compared to $SO_2$ emission in standard
E3SMv2, this configuration results in a more accurate lifetime of stratospheric sulfate aerosols following the 1991 eruption of
Mt. Pinatubo, which is consistent with observations (Baran and Foot, 1994), and the Whole Atmosphere Community Climate
Model (WACCM; Garcia et al. (2007)) with its detailed treatment of stratospheric chemistry. The model was run in a standard
low-resolution configuration of the model with approximately 1° horizontal resolution. The vertical grid consists of 72 levels
extending from 1000 hPa to 0.1 hPa, or approximately 60 km.

This model was used to generate simulation ensembles beginning on June 1st, 1991. Each ensemble member includes a
representation of the 1991 eruption of Mt. Pinatubo as an emission of 10 Tg of sulfur dioxide ($SO_2$) over 6 hours and 9
grid cells between 18 and 20 km near 15° N. The data was output as daily and monthly averages, both of which are used
here. Specifically, we utilized the 15-member "limited variability" ensembles, described in Ehrmann et al. (2024). The initial
condition for each member was obtained by applying a small, random perturbation of temperature of order $10^{-14}$ K to a base
atmospheric state. This base state was sampled from an auxiliary E3SMv2-SPA simulation run, and exhibits major climate
modes which qualitatively match the real-world conditions at the time of the 1991 Mt. Pinatubo eruption, as derived from the
Modern-Era Retrospective Analysis for Research Applications version 2 (MERRA-2; Gelaro et al. (2017)).

Though this initial state was chosen to for it's consistency with the historical scenario, the simulations are free-running (i.e.
they are not nudged toward any specific climate modes). Because the ensemble begins on June 1, 1991, the individual members
have only two weeks to diverge before the eruption occurs on June 15, 1991. This is enough time to allow for synoptic-scale
differences between members to manifest, while the large-scale circulation remains qualitatively consistent. It is in this sense
that the intra-ensemble variability is "limited", and thus the ensemble average should capture the robust climatic response to
the Mt. Pinatubo event, conditioned on the real-world initial atmospheric state. We will refer to this set of simulations as the
limited-variability volcanic ensemble (hereafter LV).

In addition, we utilize the 15-member counterfactual ensemble (hereafter CF) of Ehrmann et al. (2024), where the volcanic
sulfate injection is entirely removed. Each member of this ensemble is paired with a corresponding member from the volcanic
ensemble. These pairs are identical in their initial conditions, and identical in their evolution until June 15, when the eruption
occurs in the volcanic ensemble. After this date, the difference between the volcanic and counterfactual ensembles isolates the
net (direct and indirect) impact of the volcanic forcing.

Each pair is run for 8 years. This is enough time for the ensemble members to statistically diverge (from each other, and also
from their paired runs) such that the concept of isolating the volcanic impact becomes tenuous, and eventually meaningless.



For this reason, most of our analyses will focus on the initial 2-year period from June 1991 through June 1993, which we found to be an appropriate analysis domain for the tropics and midlatitudes (see discussions surrounding Fig. 2 and Fig. 5).

## 3  Analysis Framework

Rather than use a traditional measure of anomaly as a departure from a reference climatology, we are interested in the pair-wise difference between the LV and CF members. This approach naturally removes structures that are present in the unforced runs from consideration of the volcanic impacts, even if they are anomalous with respect to the climatology (e.g. the QBO, ENSO, and vortex states). To be clear, we will refer to this measure of the volcanic response as an "impact" rather than an "anomaly".

The TEM components of the zonal momentum budget will be used to compute impacts in large-scale wave activity, the global advection of mass, gravity wave drag, and other parameterized sources. The statistical impact expressions and TEM equation set are defined below.

### 3.1  Impact and Significance

For an $N$-member ensemble, the ensemble mean variable $x$, given the data $x^{(n)}$ from each simulation $n$, is taken independently

for both the volcanic ensemble, and the counterfactual ensemble. The counterfactual ensemble mean will be specified as $x^{\mathrm{CF}}$. Following Ehrmann et al. (2024), we define the impact on the variable $x$ as the ensemble mean of the difference between the volcanic and counterfactual data, denoting it as

$$\Delta x \equiv \frac{1}{N} \sum_{n=1}^{N} \left( x^{(n)} - x^{\mathrm{CF},(n)} \right).\tag{1}$$

Likewise, the standard deviation of the impact is

$$\mathrm{SD}^{\Delta x} = \sqrt{\frac{1}{N} \sum_{n=1}^{N} (x^{(n)} - x^{\mathrm{CF},(n)} - \Delta x)^2}.\tag{2}$$

To identify statistically significant signals in the impact, we apply a paired $t$-test, where the $t$-statistic and associated $p$-value are

$$t\text{-statistic} = \frac{\Delta x}{\mathrm{SD}^{\Delta x}/\sqrt{N}}\tag{3}$$

$$p\text{-value} = 2 \times \mathrm{CDF}(-|t\text{-statistic}|, N-1)\tag{4}$$

where CDF is the cumulative distribution function of the Student's $t$ distribution, with $N-1$ degrees of freedom, and $-|t\text{-statistic}|$ as the upper bound on the distribution integration. Our null hypothesis states that the volcanic and counterfactual ensembles are statistically indistinguishable. The alternative hypothesis is two-sided, i.e. the volcanic ensemble data is either above or below the counterfactual data, and thus a factor of two also appears in Eq. (4). Throughout the results presented in Sect. 5, we adopt a $p$-value threshold of 0.05 (95% confidence) for defining significant impacts. Confidence interval bounds of the impact





$\Delta x$ are computed as

$$\Delta x \pm t_{\text{crit}} \frac{\text{SD}^{\Delta x}}{\sqrt{N}} \tag{5}$$

where $t_{\text{crit}} = 2.145$ for $N = 15$.

All processing of the data, including zonal averaging, averaging over latitude bands, selection of vertical levels, and temporal averaging are computed at the member-level. No such processing is done directly on the ensemble means. Rather, a separate

ensemble mean, and thus separate impacts and $p$-values, will be computed for each choice of processing, per Eq. (1)–(4).

## 3.2   TEM Formulation

To efficiently diagnose the forcing on the zonal-mean flow by the residual circulation and wave activity, we employ the TEM framework originally introduced by Andrews and McIntyre (1976). First, each atmospheric variable $x$ is decomposed into a linear combination of a zonal mean $\overline{x}$ and eddy component $x'$, e.g. $u = \overline{u} + u'$ for the zonal wind $u$. The TEM momentum

equation for the evolution of $\overline{u}$ is written as the sum

$$\frac{\partial \overline{u}}{\partial t} = \frac{\partial \overline{u}}{\partial t}\bigg|_{(\overline{v}^*)} + \frac{\partial \overline{u}}{\partial t}\bigg|_{(\overline{\omega}^*)} + \frac{\partial \overline{u}}{\partial t}\bigg|_{\nabla \cdot \mathbf{F}} + \overline{X}. \tag{6}$$

We compute each of the terms on the right-hand side of Eq. (6) following the spherical-coordinate formulation specified by Gerber and Manzini (2016) (we also adopt the values of the their constants; see Appendix A2 therein). The first and second terms, which represent zonal momentum forcing by the Coriolis force and residual circulation advection, are

$$\frac{\partial \overline{u}}{\partial t}\bigg|_{(\overline{v}^*)} = \overline{v}^* \left( f - \frac{\partial \overline{u} \cos \phi}{a \cos \phi \partial \phi} \right) \tag{7}$$

$$\frac{\partial \overline{u}}{\partial t}\bigg|_{(\overline{\omega}^*)} = \overline{w}^* \frac{p}{H} \frac{\partial \overline{u}}{\partial p} \tag{8}$$

for latitude $\phi$, pressure $p$, the Earth's radius $a$, Coriolis parameter $f$, and the scale height $H = 7$ km. The meridional ($v^*$) and vertical ($w^*$) velocity components of the residual circulation are defined as

$$\overline{v}^* = \overline{v} - \frac{\partial \psi}{\partial p} \tag{9}$$

$$\overline{w}^* = -\frac{H}{p} \overline{\omega}^* = -\frac{H}{p} \left( \overline{\omega} + \frac{\partial \psi \cos \phi}{a \cos \phi \partial \phi} \right) \tag{10}$$

where $v$ is the meridional velocity, $\omega$ is the vertical pressure velocity, and

$$\psi = \frac{\overline{v'\theta'}}{\partial \overline{\theta}/\partial p} \tag{11}$$

is the eddy streamfunction, involving the potential temperature $\theta$. It will also be useful to introduce the residual circulation streamfunction $\Psi^*$,

$$\Psi^* = \frac{2\pi a \cos \phi}{g_0} \left( \int_p^0 \overline{v}^* dp \right), \tag{12}$$




were $g_0$ is the global-mean acceleration due to gravity at mean sea level. The third term on the right of Eq. (6) is the forcing of zonal momentum by resolved wave dissipation and breaking,

$$\frac{\partial \overline{u}}{\partial t}\Big|_{\nabla \cdot \mathbf{F}} = \frac{\nabla \cdot \mathbf{F}}{a\cos\phi}. \tag{13}$$

Here, $\mathbf{F}$ is known as the Eliassen-Palm (EP) flux vector, with meridional and vertical components

$$F_{(\phi)} = a\cos\phi\left(\frac{\partial \overline{u}}{\partial p}\psi - \overline{u'v'}\right) \tag{14}$$

$$F_{(p)} = a\cos\phi\left(\left[f - \frac{\partial \overline{u}\cos\phi}{a\cos\phi\partial\phi}\right]\psi - \overline{u'\omega'}\right), \tag{15}$$

and the EP-flux divergence (EPFD) is

$$\nabla \cdot \mathbf{F} = \frac{\partial F_{(\phi)}\cos\phi}{a\cos\phi\partial\phi} + \frac{\partial F_{(p)}}{\partial p}. \tag{16}$$

It is this divergence that represents the sole internal forcing of the zonal flow by transient, nonconservative waves.

Finally, the fourth term on the right of Eq. (6), $\overline{X}$, represents all unresolved forcing of $\overline{u}$, including small-scale eddies, and other parameterized sources. We will further split this into contributions from parameterized gravity waves $\overline{X}_{\text{GW}}$, and a (usually small) residual which we will generically call the diffusion $\overline{X}_d$:

$$\overline{X} \equiv \overline{X}_d - \overline{X}_{\text{GW}} \tag{17}$$

The diffusion term is inferred by taking the difference

$$\overline{X}_d = \frac{\partial \overline{u}}{\partial t} - \left(\frac{\partial \overline{u}}{\partial t}\Big|_{(\overline{v}^*)} + \frac{\partial \overline{u}}{\partial t}\Big|_{(\overline{\omega}^*)} + \frac{\partial \overline{u}}{\partial t}\Big|_{\nabla \cdot \mathbf{F}} + \overline{X}_{\text{GW}}\right), \tag{18}$$

so that the overall zonal momentum budget is closed. For the monthly-mean ensemble data, gravity wave drag outputs were available, and so it was possible to do the separation of $\overline{X}$ in Eq. (17). These outputs were not available for the daily-mean data; in that context, we keep $\overline{X}_{\text{GW}}$ on the left-hand side of Eq. (18) and can only represent the net parameterized drag $\overline{X}$ in the momentum budget.

The philosophy behind the TEM equations of motion as a diagnostic tool for wave-mean flow interaction has been articulated in several foundational works (e.g. Edmon et al. (1980); Dunkerton et al. (1981); Andrews et al. (1983)). For our purposes, the framework will allow us to identify the mechanisms that are responsible for volcanic aerosol-induced changes to the zonal flow. Specifically, by applying the methods of Sect. 3.1 to $\overline{u}$, we will identify statistically significant impacts on the zonal-mean zonal wind. We then separate the forcing of $\overline{u}$ into the contributions from the residual circulation (Eq. (7) and Eq. (8)), from resolved eddy forcing (Eq. (13)), and parameterized forcings and dissipation (Eq. (18)). By statistically comparing the closed TEM budgets of each ensemble, we may diagnose which of these dynamical mechanisms control the separation between the volcanic and counterfactual simulations at certain locations and times. The important quantities and equations used in this procedure are summarized in Table 1, which may be useful in reading the later figures.

Note that all zonal averages throughout this paper are taken by the spectral spherical-harmonic method described in Appendix B, which avoids the need to remap the data from the E3SMv2 native cubed-sphere grid to a structured latitude-longitude format.





**Table 1.** Essential quantities of the TEM framework and TEM zonal momenutm budget. The second column ("Label") refers to the labels given to the corresponding quantity in figure titles and legends throughout this work. Quantities without a label are labeled either with their name (first column), or their symbol. Units provide the SI units, though different units may appear in figures. All quantities are computed on daily-averaged simulation data, except for the GW forcing and diffusion components of the TEM budget, which are computed on monthly-averaged data.

| Quantity | Label | Symbol | Equation | Units |
|---|---|---|---|---|
| **TEM quantities**: | | | | |
| meridional residual velocity | — | $v^*$ | Eq. (9) | m s$^{-1}$ |
| vertical residual velocity | — | $w^*$ | Eq. (10) | m s$^{-1}$ |
| residual circulation streamfunction | RC streamfunction | $\Psi^*$ | Eq. (11) | kg s$^{-1}$ |
| EP flux vector | EP flux | $\left[F_{(\phi)}, F_{(p)}\right]$ | Eq. (14, 15) | m$^3$ s$^{-2}$, m$^2$ s$^{-2}$ Pa |
| EP flux divergence | EPFD | $\nabla \cdot \mathbf{F}$ | Eq. (16) | m$^2$ s$^{-2}$ |
| | | | | |
| **TEM budget (zonal-mean zonal wind forcings)**: | | | | |
| net forcing | — | $\frac{\partial \overline{u}}{\partial t}$ | Eq. (6) | m s$^{-2}$ |
| meridional residual velocity advection | $v^*$ forcing | $\left.\frac{\partial \overline{u}}{\partial t}\right|_{(v^*)}$ | Eq. (7) | m s$^{-2}$ |
| vertical residual velocity advection and Coriolis force | $w^*$ forcing | $\left.\frac{\partial \overline{u}}{\partial t}\right|_{(\omega^*)}$ | Eq. (8) | m s$^{-2}$ |
| total residual circulation forcing | RC forcing | — | Eq. (7) + Eq. (8) | m s$^{-2}$ |
| resolved eddy forcing (large-scale wave drag) | EPFD forcing | $\left.\frac{\partial \overline{u}}{\partial t}\right|_{\nabla \cdot \mathbf{F}}$ | Eq. (13) | m s$^{-2}$ |
| unresolved (parameterized and implicit) forcings | unresolved | $\overline{X}$ | Eq. (17) | m s$^{-2}$ |
| gravity wave drag | GW forcing | $\overline{X}_{\mathrm{GW}}$ | Eq. (17) | m s$^{-2}$ |
| unresolved forcings beyond gravity waves | diffusion | $\overline{X}_{\mathrm{d}}$ | Eq. (18) | m s$^{-2}$ |

In computing Eq. (18), the total tendency $\partial \overline{u}/\partial t$ was not available to us as a model output in the simulation ensembles, and so it is constructed from the daily-mean $\overline{u}$ by a first-order finite forward difference. Meridional and vertical derivatives are computed with a second-order centered finite-difference. See Appendix A for details.

## 4 Reference State

### 4.1 Midlatitudes

In order to establish the behavior of the unforced simulations during the Pinatubo period, Fig. 1(a)–(d) shows the seasonal zonal-mean zonal-wind structure for a single year of the CF ensemble mean, spanning June 1991 to June 1992. The stratospheric winds in the E3SMv2-SPA model appear to be broadly consistent with climatological averages deduced from the ERA5 reanalyses dataset (Fig. 2 in Butchart (2022)), with the SH winter-time polar vortex reaching maximum wind speeds in excess of 80 m s$^{-1}$ toward the model top and near 60°S, and the NH polar vortex reaching speeds above 50 m s$^{-1}$ near 60°N. In each





summer hemisphere, easterlies peak at 30–40 m s$^{-1}$ near 20°N and 20°S. These wind speeds are slightly low for the SH, and slightly high for the NH with respect to the reanalysis climatology, but are well within the range of observed variability (Fig. 8 in Butchart (2022)). There is similar accuracy in the tropospheric jets.

In addition, EP flux vectors and contours of the residual circulation streamfunction are plotted over the seasonal winds in Fig. 1(a)–(d). Negative values in streamfunction (dashed contours) indicate counter-clockwise circulation the meridional plane, and vice-versa for positive values (solid contours). The zero-line between these regions is drawn in bold. Qualitatively, this shows equator-to-pole overturning cells in the lower stratosphere and below, and a single pole-to-pole circulation from summer-to-winter hemisphere during the solstitial seasons. These two regimes represent the advective components of the well-known shallow and deep branches of the BDC, respectively (Birner and Bönisch, 2011; Butchart, 2014).

The EP flux vectors show the relative magnitude and direction of resolved wave propagation in the meridional plane (Edmon et al., 1980; Andrews, 1987). In the summer hemisphere, wave activity forced by surface processes is largely constrained below the tropopause (e.g. Fig. 1(a), NH), since the stratospheric easterlies aloft prevent the vertical propagation of Rossby waves (Charney and Drazin, 1961). During the following autumn, the development of upper-level westerlies acts as a valve for wave propagation into the stratosphere (e.g. Fig. 1(b), NH). This persists through the winter, when stratospheric wave activity (and thus, zonal-wind variability) is at its highest (e.g. Fig. 1(c), NH).

The seasonal TEM momentum balance is shown in Fig. 1(e)–(h) at 3 hPa (winter vortex core) and 30 hPa (lower vortex edge). Forcing by the EPFD, the residual circulation, parameterized gravity wave drag, subgrid diffusion, and their sum are plotted as functions of latitude, along with the CF ensemble mean $\overline{u}$. Scaling of the vertical axis is unique in each panel. This reveals the familiar result (e.g. Andrews et al. (1983)) that the net tendency $\partial \overline{u}/\partial t$ in the stratosphere is due to a relatively small imbalance between negative wave-driven forcing (dissipation and breaking of resolved large-scale waves and parameterized gravity waves), and positive forcing by the residual circulation (Coriolis torque and advection of momentum).

In the summer hemisphere, both the wave-driven and residual circulation forcing magnitudes are low, and the compensation between them is nearly complete (e.g. Fig. 1(e), NH). The result is mild easterly wind speeds with low variability. In the winter hemisphere (e.g. Fig. 1(g), NH), approximate balance between the forcing terms remains, but the relative contributions are much larger. The result is a strong polar jet. The equinoctial seasons serve as the transitions between the two quasi-steady states found in the summer and winter stratosphere, at which time the force balance must break, and the total mean forcing on $\overline{u}$ departs from zero (Fig. 1(f,h)). In the autumnal hemisphere, net positive forcing arises as the polar vortex is spun-up by the Coriolis torque of the strengthening meridional residual velocity, while in the vernal hemisphere, net negative forcing results as wave drag erodes the vortex.

Note that in both winter hemispheres at 3 hPa (Fig. 1(e), SH and Fig. 1(g), NH) there tends to be a sign change in the EPFD on each side of the polar vortex, such that wave breaking and dissipation (EPFD< 0) occurs equatorward, in the surf zone (McIntyre and Palmer, 1984), and wave divergence (EPFD> 0) occurs poleward. This is qualitatively consistent with some reanalysis climatology results, e.g. ERA-Interim (Díaz-Durán et al., 2017). The parameterized gravity wave drag (yellow curve) in the model tends to exhibit the opposite behavior.



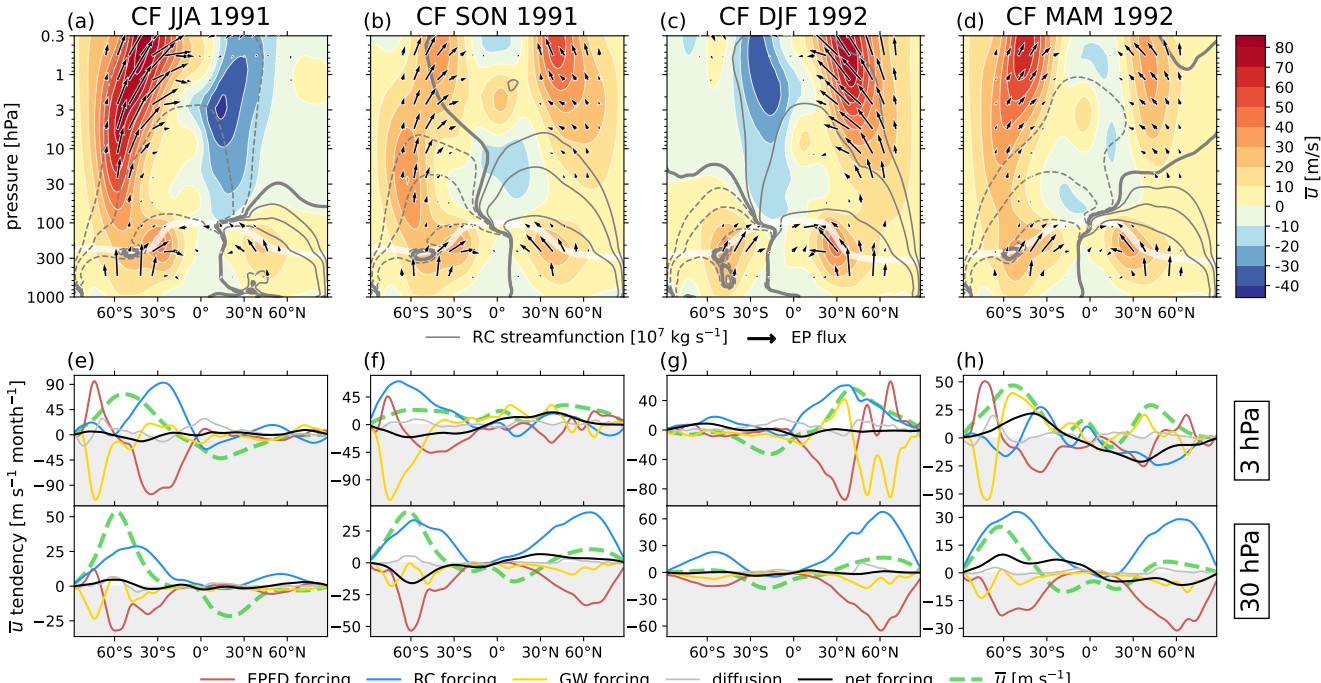

**Figure 1.** Seasonal zonal wind and TEM momentum balance for the first year of the CF ensemble mean. **(a–d)** $\overline{u}$ averaged over 3-month seasons in contours of 10 m s$^{-1}$, with EP flux vectors, gray $\Psi^*$ contours, and the tropopause overplotted as a thick, faint white contour. The EP flux vectors are scaled following Jucker (2021). The $\Psi^*$ contours are drawn at 30, 100, and 500 in units of $10^7$ kg s$^{-1}$ on each side of zero (bold contour). Negative contours are dashed. **(e–h)** seasonal contributions to $\partial\overline{u}/\partial t$ in [m s$^{-1}$ month$^{-1}$] from the EPFD (red solid), the residual circulation (blue solid), gravity waves (gold solid), and diffusion (thin gray solid) as functions of latitude at 3 hPa (top panels) and 30 hPa (bottom panels). Also shown is the net tendency (black solid), as well as $\overline{u}$ itself (thick green dashed) in [m s$^{-1}$]. The negative tendency (and wind speed) domain is shaded in light gray. Note that each panel of (e)–(h) has unique scaling of the vertical axis, such that all curves are contained within the plotting region.

At 30 hPa, well below the vortex peak, there is broad negative EPFD from the midlatitudes to the poles. This westward wave driving is about a factor of 2 larger in magnitude in the NH than the SH due to the continental land masses—an effect which might have the capacity to trigger a breakdown of the vortex (Baldwin et al., 2021) in a different realization.

## 4.2 Tropics

The tropical zonal-mean zonal wind averaged over 5°S–5°N in the CF ensemble mean is shown in Fig. 2 for the full 8-year simulated time period. In Fig. 2(a), the initial QBO phase is easterly, and its first cycle has a period of approximately 24 months with peak winds of about $-10$ m s$^{-1}$ and 7 m s$^{-1}$ in January of 1991 and 1992 respectively, at 30 hPa in the ensemble mean. This is weak by a factor of 2 to 3 compared to ERA5, which is a known bias of E3SMv2 (Yu et al., 2024).



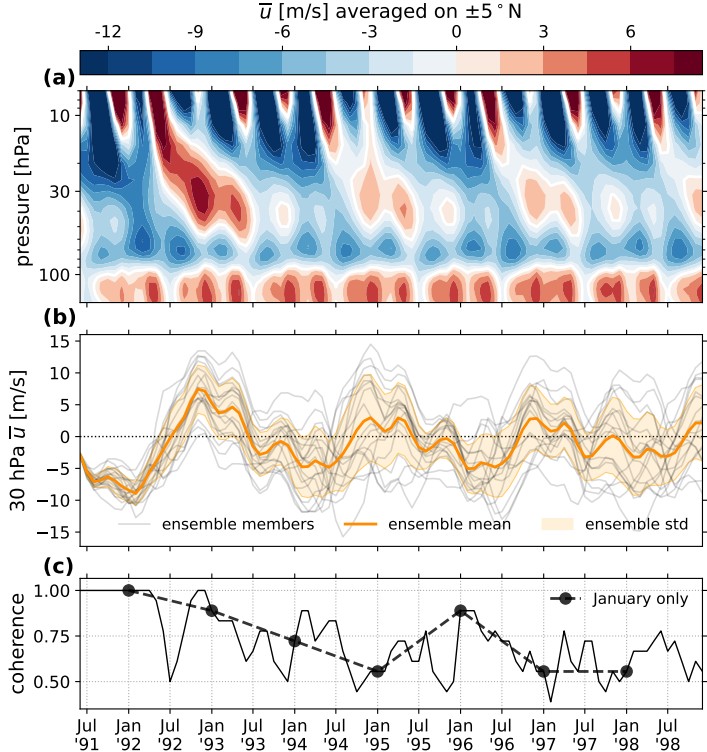

**Figure 2.** The QBO shown as $\overline{u}$ averaged over [5°S, 5°N] for the CF ensemble mean. **(a)** time–pressure plane centered on the lower stratosphere. **(b)** zonal wind time series at 30 hPa. The ensemble mean and standard deviation in $\overline{u}$ is shown as a bold orange line and light orange shading, respectively. Individual ensemble members are shown as faint gray lines, and the zero-line is dotted. **(c)** the ensemble coherence, defined as the fraction of ensemble members in agreement with the sign of the ensemble mean in panel (b). A bold dashed line with points displays the coherence evaluated only at each January.

In addition to a weak amplitude compared to reanalyses, the semi-annual oscillation (SAO) penetrates deeper into the stratosphere, and the descent of QBO phases is typically hastened in E3SMv2, resulting in a shortened period at fixed pressure.

Yu et al. (2024) showed that the spectral density of the QBO computed over 1985–2015 at 20 hPa in E3SMv2 is approximately uniform in power from 20 to 30 months in period, with no distinct peak. For our realizations in E3SMv2-SPA, the QBO manifests with a period of about 24 months. This causes a seasonal phase lock, where maxima and minima in QBO wind speeds occur during alternating boreal winters.

Following the first cycle, the QBO signal in the ensemble mean weakens further, by an additional factor of ∼2. Most of this

weakness is explained by the increasing intra-ensemble spread, as the members diverge from their common initial condition, demonstrated at 30 hPa in Fig. 2(b). Alternatively, Fig. 2(c) shows the 30 hPa ensemble "coherence", or the fraction of the members which agree in sign with the ensemble mean. The coherence is variable, unsurprisingly dropping to 50% whenever $\overline{u}$



crosses zero. More useful is the January coherence (bold dashed line and points) which, because of the seasonal phase lock of the QBO in the model, shows the ensemble phase agreement during each easterly and westerly maximum. We observe that the

January coherence drops below 80% near July of 1993, and so for the remainder of this study we will focus only on this early 2-year period. In the midlatitudes, we should expect the ensemble coherence to fall off faster, though we will still conduct the analysis over this same period.

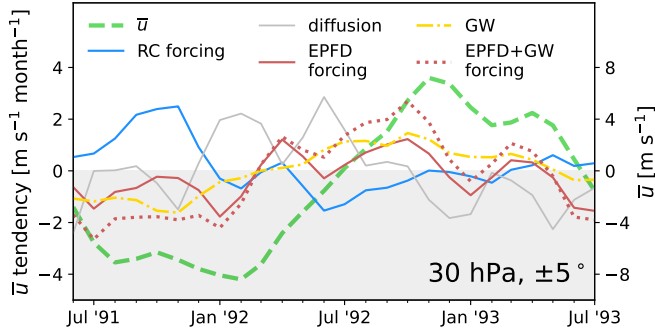

**Figure 3.** The TEM momentum balance of the QBO for the first two years of the CF ensemble mean, averaged over [5°S, 5°N] at 30 hPa. Shown are the contributions to $\partial \overline{u}/\partial t$ in [m s$^{-1}$ month$^{-1}$] from the EPFD (red solid), gravity waves (gold dotted), the combined EPFD and gravity waves (red dotted), the residual circulation (blue solid), and diffusion (thin gray solid) as functions of latitude at 3 hPa (top panels) and 30 hPa (bottom panels). Also shown is $\overline{u}$ itself (thick green dashed) in [m s$^{-1}$], values read on the left vertical axis. The negative tendency or wind speed domain is shaded in light gray.

The TEM balance describing the QBO evolution at 30 hPa over the initial 2-year period is shown in Fig. 3. Curves show the monthly-mean time series of the forcing on $\overline{u}$ by the EPFD and gravity waves, the residual circulation, and diffusion (this

is the same data as in Fig. 1(e)–(h) before seasonal averaging, but with the net forcing omitted). The sum of the EPFD and gravity wave drag is also drawn (red dotted line), which shows the usual result that wave-driving is the dominant process controlling the QBO phase descent (Baldwin et al., 2001). However, momentum transport by vertically propagating tropical waves is actually much larger than the net motion of the QBO would suggest (peaking at ∼3 m s$^{-1}$month$^{-1}$ in NH autumn of 1992, while the net tendency is nearer to ∼1.3 m s$^{-1}$month$^{-1}$), since it is partially canceled by the residual circulation (blue

solid line), which hinders the descending phase (Dunkerton, 1997). As the Coriolis force is negligible at these latitudes, and $v^*$ is small, this forcing can be interpreted primarily as a diabatic effect, where upwelling of momentum by $w^*$ is in control (see steamfunction contours in Fig. 1). The result is positive (negative) forcing by the residual circulation in westerly (easterly) shear zones (as in e.g. Brown et al. (2023)).

## 5   Volcanic Impact Results

With an understanding of the seasonal TEM balance in the CF ensemble established, we now investigate disruptions to this balance by the Pinatubo aerosol forcing. Figure 4 shows the monthly-averaged global $\Delta\overline{u}$ for select months in the autumn of




1991, winter of 1992, and autumn of 1992 (panels (a)–(c)), as well as meridionally-resolved impacts over 70°S–70°N as a function of time at 10 hPa and 30 hPa (panels (d, e)). On all panels, the counterfactual winds are drawn in black contours every 10 m s$^{-1}$. Regions where the statistical significance of the impact is above 95% are enclosed by a bold white contour, and
regions below 95% are marked with white hatching.

Midlatitude and tropical impacts are discussed in Sect. 5.1 and 5.2, respectively. The strategy is to (1) identify localized regions of significant $\Delta\overline{u}$ in space and time, (2) over the each identified region, decompose $\Delta(\partial\overline{u}/\partial t)$ into the TEM form of Eq. (6), and (3) analyze the imbalance between the impact of the TEM terms. If the result of this process is a set of TEM impacts which are themselves statistically significant, then we will consider the specific mechanism in control of the associated
$\Delta\overline{u}$ to have been found.

In the midlatitudes, this process amounts to answering the question; if thermal wind balance is to be approximately maintained after a source of external forcing is introduced, then what processes govern the required adjustments to the zonal-wind to that end? From a TEM perspective, it might be assumed that the residual circulation and large-scale wave drag conspire in this purpose, but as we will show, the quantitative details of this balance change notably with at least season and latitude.

This type of analysis has precedent in the literature, most notably Bittner et al. (2016) and Toohey et al. (2014), but we are not aware of another work on volcanic aerosol forcing that attempts to give a complete accounting of the closed TEM budget in this way. We take some inspiration from similar closed-budget analyses that have been done for climate trends in tracer distributions, e.g. Abalos et al. (2013, 2017, 2020).

### 5.1 Impacts in the Surf Zone & Polar Vortex

Figure 4(d, e) shows that until 2 years post-eruption, essentially all stratospheric $\overline{u}$ impacts outside of the tropics are westerly in nature, consistent with aerosol-driven enhancements of the meridional temperature gradient. These features correspond to either zonal acceleration or deceleration with respect to the reference, depending on the season.

The significant response begins near 10 hPa in both hemispheres (Fig. 4(d)). In the northern hemisphere, a westerly impact of ~3 m s$^{-1}$ develops near 30°N, decelerating easterly winds (black dashed contours) from July through the end of the NH
summer of the eruption. This impact moves poleward toward 60°N throughout autumn, acting to hasten the spin-up of the polar vortex along its equatorward edge, before becoming insignificant in early November. The vertical structure averaged over October 1991 is given in Fig. 4(a), which suggests that this feature is perhaps part of a subtle equatorward shift of the vortex, given the accompanying easterly impact near the vortex core above 1 hPa. We will refer to this impact signature as an the "summer response" (SR). Note that a similar summer response occurs again from July to October of 1992 near 30°N, as well
as in the southern hemisphere between December and April 1992, albeit at lower latitudes.

Following the 1991 SR is a stronger westerly impact of up to ~6 m s$^{-1}$ occurring at both 10 and 30 hPa near 40°N, between January and May 1992 (Fig. 4(d, e)). This impact serves to accelerate westerlies in the surf zone (equatorward edge of the polar vortex). Figure 4(b) shows the vertical structure averaged over February of 1992, in which we again see easterly impacts near and poleward of the vortex core, indicative of equatorward vortex shift (though it is insignificant). We will refer to this
impact as the "winter response" (WR), and we identify this response most closely with the claims of an "accelerated vortex





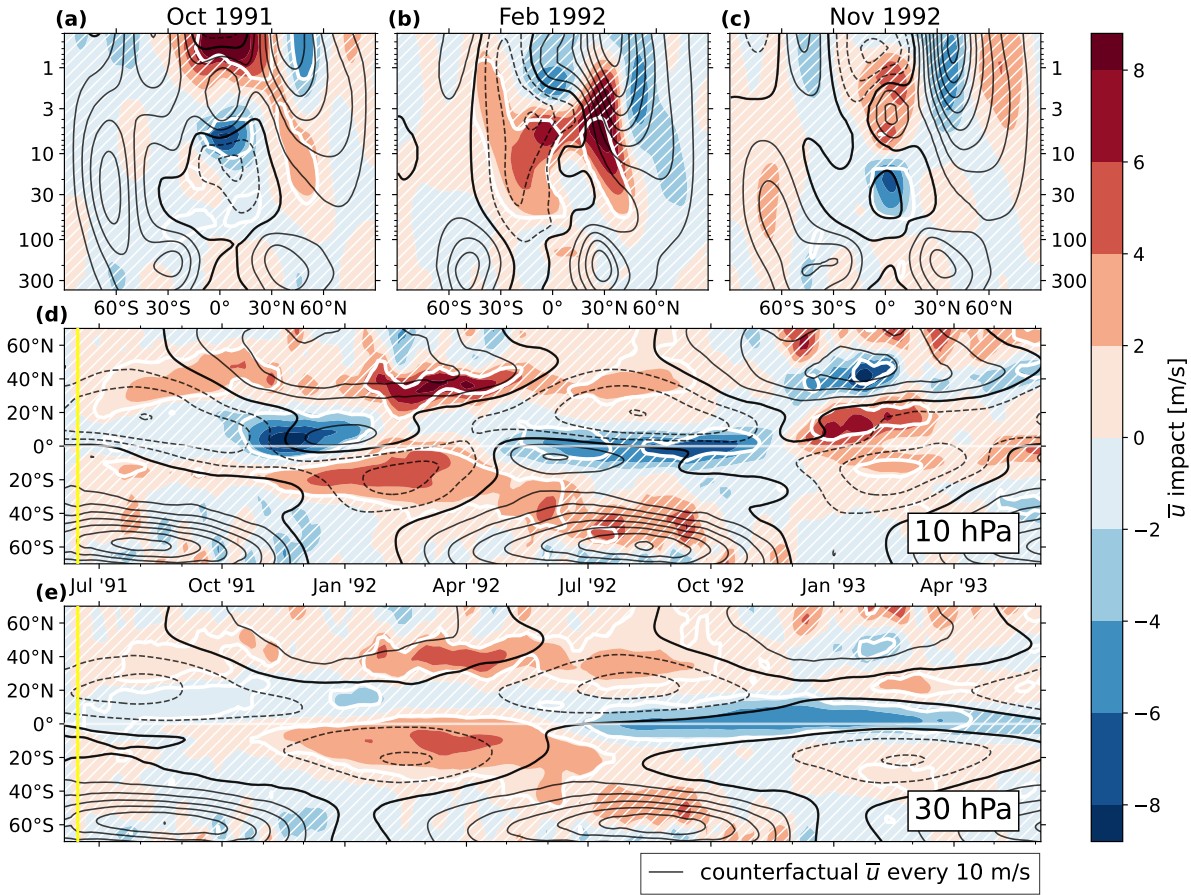

**Figure 4.** The ensemble-mean $\Delta\overline{u}$ as filled contours (colorscale), with the CF ensemble-mean $\overline{u}$ overplotted as black contours, drawn every 10 m s$^{-1}$, with negative contours dashed and the zero-line in bold. A bold white contour is drawn at 95% significance, with the $\Delta\overline{u}$ $p$-value computed as in Sect. 3.1. Regions of insignificance are filled with white hatching. The upper three panels show the latitude-pressure plane for time averages over **(a)** October 1991, **(b)** February 1992, and **(c)** November 1992. The lower two panels show the time-latitude plane for pressure levels at **(d)** 10 hPa, and **(e)** 30 hPa. In panels (d) and (e), a vertical yellow line shows the time of eruption, and a faint white line is drawn on the equator.

region" that are well-represented in the literature, as discussed in Sect. 1. A similar (but less significant) repose also occurs in the southern hemisphere near 50°S between June and October 1992.

The summer and winter responses are summarized in Fig. 5, which shows the separation of the LV and CF ensembles in one-dimension as $\overline{u}$ and $\Delta\overline{u}$ at 20 hPa, averaged over 30–50°N. Also plotted are the 95% confidence intervals on the ensemble means and the impact, as well as a curves at two standard deviations. In the lower panel, the confidence interval is shaded in green where the impact is significant, which clearly shows the 1991 SR, 1992 WR, and 1992 SR. This view illustrates that the relatively strong winter-time stratospheric variability in the vortex region sets a kind of lower-bound on the forcing magnitude



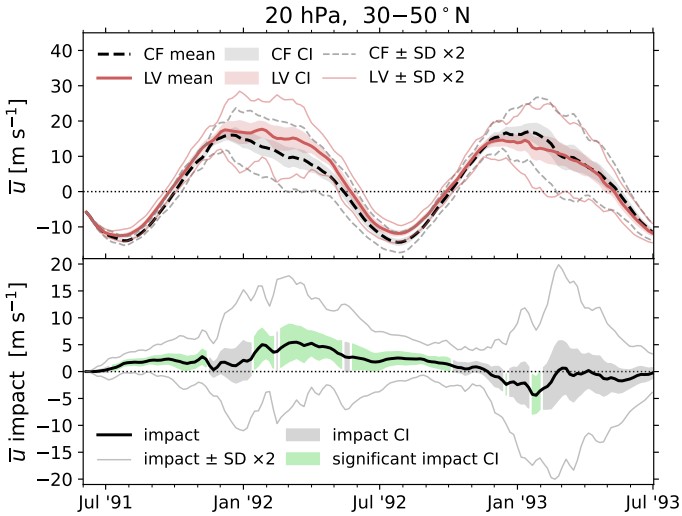

**Figure 5.** Time series of the CF ensemble-mean $\overline{u}$ and LV ensemble-mean $\overline{u}$ (upper panel) as well as the ensemble-mean $\Delta\overline{u}$ (lower panel) at 20 hPa, averaged over 30–50°N. In both panels, shaded bands give the confidence intervals, as computed in the way of Sect. 3.1, and thin lines give two standard deviations on each side of the mean. In the lower panel, the confidence interval is shaded in light green where the impact is statistically significant at the 95% level.

that is required to illicit a significant response, which our simulations are only just managing to overcome. In the summer-time quiescent stratosphere, on the other hand, the noise floor is much lower, and so even small impacts can be detected. This is

consistent with the findings of previous work that has demonstrated a statistically weak response from Pinatubo-like forcing, even for much larger ensemble sizes (Bittner et al., 2016).

For the remainder of this section, our analysis will be restricted to the northern hemisphere, since those impacts occur earlier and with more significance, and have more representation in the literature. There is some suggestion of an easterly impact in boreal winter of 1993 (Fig. 4(d)), but it is largely insignificant. Hence, we will also restrict the analysis of this section to the

the first 12–16 months post-eruption.

### 5.1.1   TEM Balance of the Surf Zone & Polar Vortex

We now investigate the impacts to the TEM balance associated with both the summer and winter midlatitude responses. Figure 6 picks out the 30–50° latitude band at 10 hPa from Fig. 4(d), and reproduces it in its panel (a). Panel (b) shows the vertically-resolved meridional average over this band. Over this domain, in addition to $\Delta\overline{u}$, we also obtain the impact (as Eq. (1)) of the

net tendency $\partial\overline{u}/\partial t$, the large-scale wave drag (Eq. 13), the residual circulation forcing (Eq. (7) + Eq. (8)), gravity wave drag, and diffusion (Eq. (18)). We then define time windows which span the 1991 SR, 1992 WR, and 1992 SR. Next, we individually



integrate the TEM forcing terms from the left (time $t_0$) to the right (time $t_1$) ends of each window, all of which are given a common initial condition of $\overline{u}(t_0)$ as evaluated in the LV ensemble mean.

In this way, the sum of the integrated TEM impacts are identical to the integration of the $\Delta(\partial \overline{u}/\partial t)$, which in turn is identical to the observed $\Delta \overline{u}$. This procedure is described in detail in Appendix A, and the result is shown in Fig. 6(c)–(e). The intention is to visualize the *accumulated* contribution to the observed $\Delta \overline{u}$ by each of the large-scale TEM processes. In other words, each curve shown in panels (c)–(e) show the anomalous evolution of the wind in absence of all other forcings, given the shared initial condition. This is preferable to showing the forcing impacts themselves, since those often change sign on small temporal scales, and thus the noise would visually dominate over the meaningful trend.

Each integration time window is shown as a highlighted yellow-green strip in panel (b) at 10 hPa. The integrated tendency contributions are shown in panel (c) for the 1991 SR, panel (d) for the 1992 WR, and panel (e) for the 1992 SR. It is immediately apparent that while these westerly impacts appear similar from a thermal-wind perspective, they are in fact brought about by different mechanisms.

The summer-time responses of 1991 and 1992 (panels (c) and (e)), are characterized by a deficit of large-scale wave forcing, and a surplus of forcing by the residual circulation. While the magnitude of impact on each of these effects is comparable, it is the residual-circulation forcing that is favored in the imbalance, and is ultimately responsible for the manifest $\Delta \overline{u}$. Specifically, the forcing is almost entirely due to the impact on the meridional component, $\Delta v^*$, and can be interpreted primarily as a Coriolis-driven acceleration, while $\Delta w^*$ is very small (as expected in the extratropics). Toward the end of the integration time windows, it appears that an increasingly negative $\Delta(\text{EPFD})$ eventually dominates and brings the summer responses to an end. Impacts on gravity wave and diffusion forcing play a more minor role, and have opposite signs in 1991 and 1992.

On the other hand, the winter-time response is primarily wave-driven (panel (d)). From January to April 1992, $\Delta(\text{EPFD})$ and $\Delta v^*$ are both large (nearly an order of magnitude larger than the previous summer), though the imbalance favors the former. This drives a net westerly acceleration. By spring of 1992, the TEM forcings are re-aligning with the reference runs (the integrated tendency curves are flattening), and the restored balance brings the winter response to an end.

In order to clarify the nature of these mechanisms of impact, Fig. 7, Fig. 8, and Fig. 9 show the complete TEM budget for the zonal-wind impact $\Delta \overline{u}$ in the meridional plane for select monthly means during the 1992 WR, 1991 SR, and the 1992 SR, respectively. In each of these figures, panel (a) shows $\Delta \overline{u}$, $\Delta \overline{u}$ significance, and $\overline{u}^{\text{CF}}$, in the way of Fig. 4. Panels (b)–(e) then show the impacts of the TEM forcing terms, and their significance, while panels (f)–(i) show the reference forcings from the CF ensemble. In panels (f) and (b), the CF EP flux vectors and their impacts are shown, respectively. Likewise, in panels (g) and (c), the CF $\Psi^*$ tangent vectors and their impacts are shown, which have been scaled using a method detailed in Appendix C. In all panels (b)–(i), the significance contours from panel (a) are reproduced in black, for spatial reference. The monthly averaging periods were informed by Fig. 6, chosen such that we analyze times when the forcing impacts are strong (i.e. integrated curves in Fig. 6(c)–(e) are steep).

Figure 7 reiterates that the winter-time acceleration of westerly winds at the equatorward vortex edge are primarily a consequence of perturbed wave activity. Panel (b) shows a statistically significant surplus of large-scale wave drag (negative forcing impact) at the lower, equatorward edge of the region of significance in $\Delta \overline{u}$, and a deficit of drag aloft (positive forcing impact).



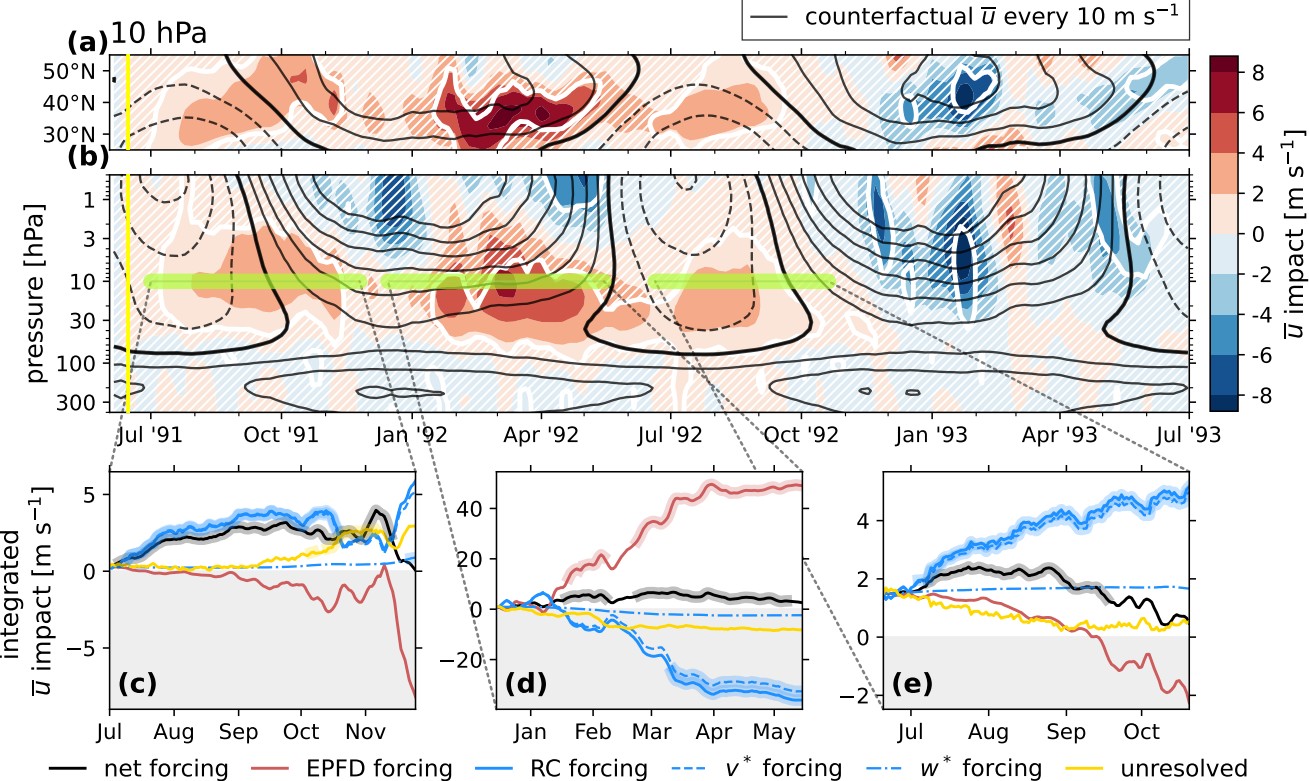

**Figure 6.** Vertical structure of the midlatitude zonal-wind impacts, and the TEM balance impact at 10 hPa. Panels **(a)** and **(b)** show the ensemble-mean $\Delta\overline{u}$ as filled contours (colorscale), with the CF ensemble mean $\overline{u}$ overplotted as black contours, drawn every 10 m s$^{-1}$, with negative contours dashed and the zero-line in bold. A bold white contour is drawn at 95% significance, with the $\Delta\overline{u}$ $p$-value computed as in Sect. 3.1. Regions of insignificance are filled with white hatching. **(a)** shows the latitude band from 25°N to 55°N at 10 hPa, reproduced from Fig. 4(b). **(b)** shows the meridional mean over 30–50°N, from 400 to 0.5 hPa, for two years following the eruption (which is indicated with a vertical yellow line). In panels **(c–e)**, curves show the integrated forcing imapct by the EPFD (red solid), by $w^*$ advection (dash-dotted blue), by $v^*$ advection and the associated Coriolis force (dashed blue), and by $\overline{X} = \overline{X}_{\mathrm{GW}} + \overline{X}_d$ (yellow solid). Also shown is the cumulative residual velocity forcing (blue solid), and the total forcing (black solid). The curves are backed by a thick shading of a matching color where they are statistically significant. Each term has units of m s$^{-1}$ after time integration. The total forcing curve (black solid) matches the data on the colorscale in the corresponding highlighted region of panel (b). These data are computed according to Appendix A. The negative domain is shaded.

Comparing the vectors of panel (b) and panel (f) reveals the cause for this effect. Relative to the CF ensemble mean, vertical and equatorward wave propagation is decreased near 50°N and 10 hPa, while it is enhanced near 30°N and 30 hPa. This result is exactly the "wave deflection" mechanism proposed by Bittner et al. (2016), where enhanced westerlies near 30°N alter the background condition for wave propagation in the surf zone, serving to steer Rossby waves away from the vortex. With those

420



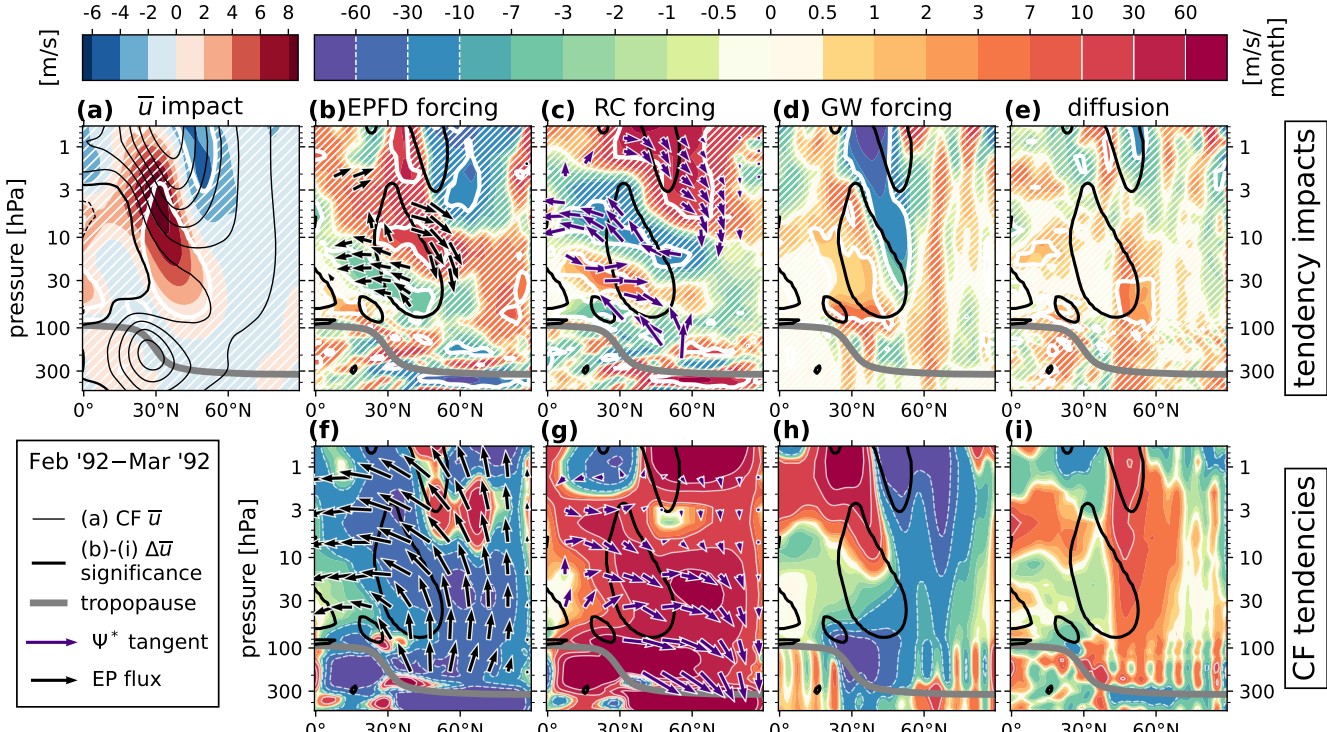

**Figure 7.** The complete TEM budget in the northern hemisphere meridional plane from 500 hPa to 0.5 hPa, averaged from February 1 1992 to March 1 1992. **(a)** the ensemble-mean $\Delta\overline{u}$ as filled contours (red/blue color scale), with the CF ensemble-mean $\overline{u}$ overplotted as black contours, drawn every 10 m s$^{-1}$, with negative contours dashed and the zero-line in bold. For every other vertical pair of panels, the top and bottom panels show the ensemble-mean impact and CF ensemble-mean of a variable as filled contours (rainbow colorscale). Specifically, **(b)** and **(f)** shows the EPFD forcing (Eq. (13)). Black vectors drawn in these panels show the significant ensemble-mean impact and the CF ensemble-mean EP flux vectors (Eq. (14, 15)), respectively, scaled according to Jucker (2021). The vector lengths are additionally log-scaled equally in length, in order to effectively visualize the vector directions. **(c)** and **(g)** show the residual-circulation forcing (Eq. (7) + Eq. (8)). Purple vectors drawn in these panels show the significant ensemble-mean impact and the CF ensemble-mean $\Psi^*$ tangent vectors (Eq. (11)), respectively, scaled according to Appendix C. For both $\Psi^*$ and the EP flux, vectors near and below the tropopause are removed. **(d)** and **(h)** show the gravity-wave forcing. **(e)** and **(i)** show the forcing by diffusion (Eq. (18)). On all panels, regions of 95% statistical significance are enclosed with a bold white contour for the variable plotted on the colorscale, and regions of insignificance are filled with white hatching. For spatial reference, the significance contour of the net impact $\Delta\overline{u}$ (bold white contour in panel (a)) is reproduced as a solid black contour in all other panels. Also plotted on all panels is the tropopause, as a bold grey curve. In the rainbow colorscale used for the tendencies, thin solid (dashed) white contours are drawn between all values larger than positive (negative) 10 m s$^{-1}$month$^{-1}$ in magnitude.

waves breaking at relatively lower latitudes, the vortex experiences less drag, and higher wind speeds. In our simulations, this effect does not extend to the vortex core, but the mechanism appears to be the same.





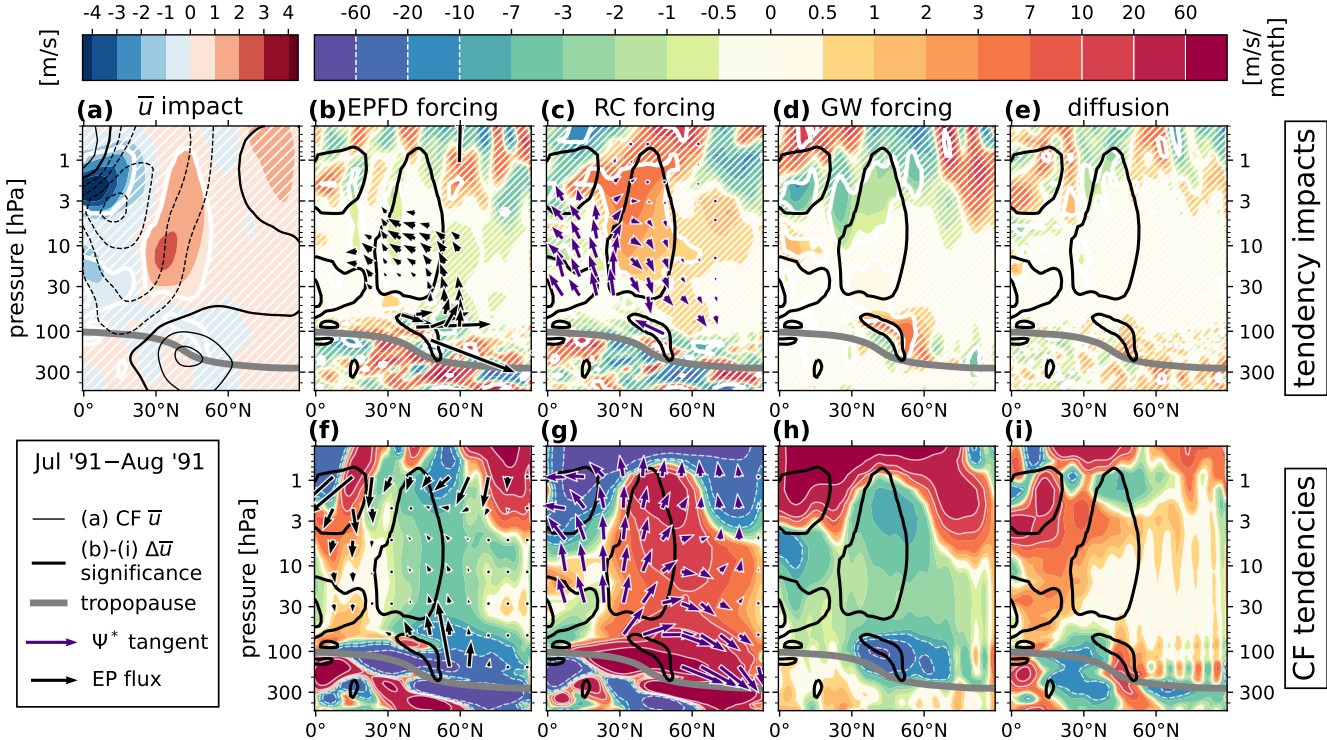

**Figure 8.** The complete TEM budget in the northern hemisphere meridional plane from 500 hPa to 0.5 hPa, averaged from July 1 1991 to August 1 1991. See the caption of Fig. 7 for full panel descriptions, considering the following modifications: In panel (**b**), the EP flux vectors are scaled in length to an order of magnitude smaller than the CF vectors in panel (**f**), and both set of vectors are linearly-scaled in length rather than log-scaled. In panels (**c**) the $\Psi^*$ tangent vectors are scaled to two orders of magnitude smaller than the CF vectors in panel (**g**) in length, and both sets of vectors are log-scaled.

At the same time, the residual circulation contribution to the forcing impact opposes the CF condition at 10 hPa, but reinforces it at 30 hPa, and near the model top. Comparing the vectors of panel (c) and panel (g), this appears to simultaneously suggest (1) a localized deceleration of the midlatitude shallow branch and (2) an acceleration of the deep branch of the advective BDC. In addition, panels (d,e,h,i) show that while positive impacts to the forcing by gravity wave drag and diffusion are present, they are minor contributors to the net response.

Figures 8 and 9 show the analogous results for the 1991 SR and 1992 SR, which reiterate that the summer-time westerly impacts are primarily a consequence of enhanced Coriolis torque from positive impacts on $v^*$. This is demonstrated in panels (c), which shows a strengthened poleward meridional circulation (vector fields), and an associated positive forcing impact over the entire region of $\Delta\overline{u}$ significance. These anomalous circulation cells project strongly onto the counterfactual residual circulation (panels (g)) at 10 hPa and below, which we interpret as an acceleration of the shallow branch of the advective BDC. During the summer of 1991, this acceleration is limited to the subtropics, but has expanded to the pole by the summer of 1992.



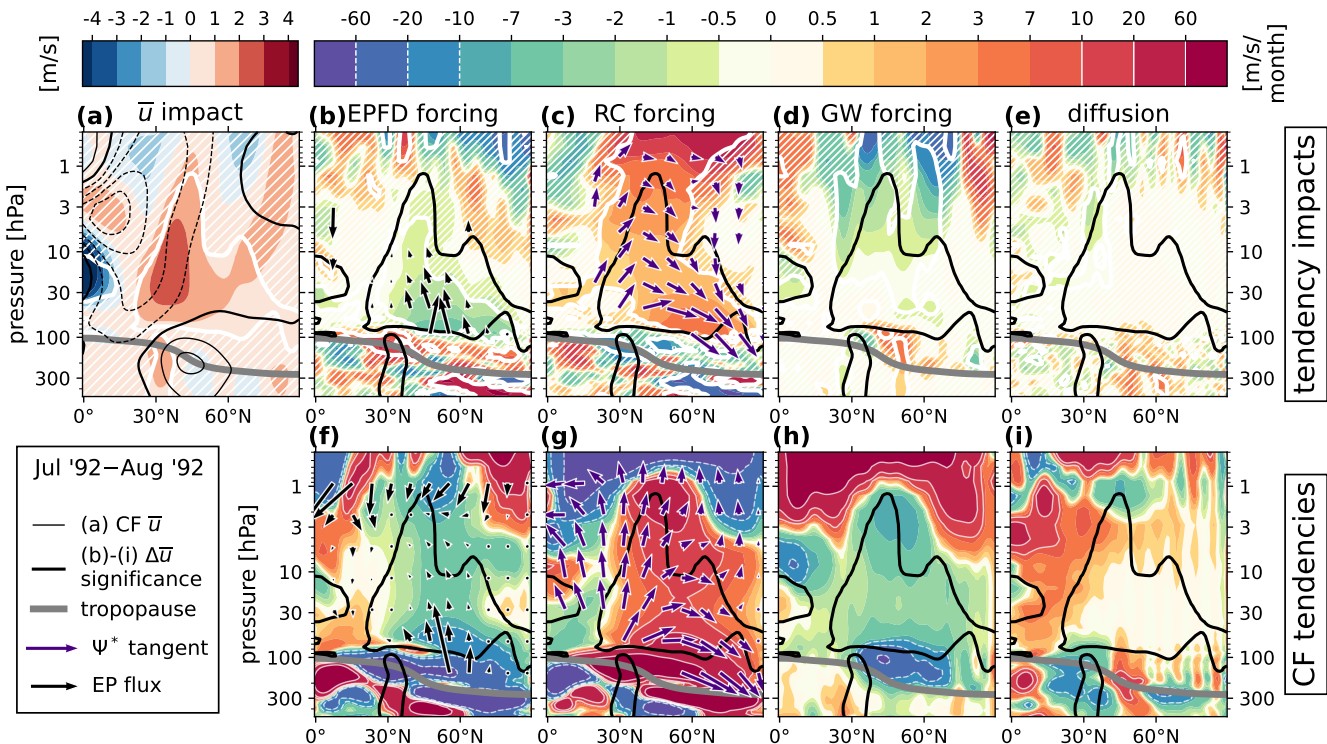

**Figure 9.** The complete TEM budget in the northern hemisphere meridional plane from 500 hPa to 0.5 hPa, averaged from July 1 1992 to August 1 1992. See the caption of Fig. 7 for full panel descriptions, considering the following modifications: In panel **(b)**, the EP flux vectors are scaled in length to an order of magnitude smaller than the CF vectors in panel **(f)**, and both set of vectors are linearly-scaled in length rather than log-scaled. In panels **(c)** the $\Psi^*$ tangent vectors are scaled to two orders of magnitude smaller than the CF vectors in panel **(g)** in length, and both sets of vectors are log-scaled.

Meanwhile, in panels (b,f,d,h), we see that impacts to both resolved and unresolved wave drag are either weak, or oppose the sign of the net westerly impact. In the case of resolved Rossby waves, the EP flux vector impacts shown in panels (b) indicate that expanded regions of westerly $\overline{u}$ near 30 hPa and below are allowing vertical wave propagation, which would otherwise have been prevented by the presence of easterlies (especially in Figure 9).

## 5.2 Impacts in the Tropics

Figure 4(d, e) shows that the significant impacts to the zonal circulation in the tropics arise and persist over a much longer timescale than those in the midlatitudes. The first of these occur in the NH autumn 1991 at 10 hPa (Fig. 4(d)) as strong easterly, followed by westerly impacts between 20°S–20°N. The October 1991 mean given in Fig. 4(a) shows that this feature is paired with a complimentary tropical impact above 1 hPa, which may be understood as a lag in the descent of the westerly phase of




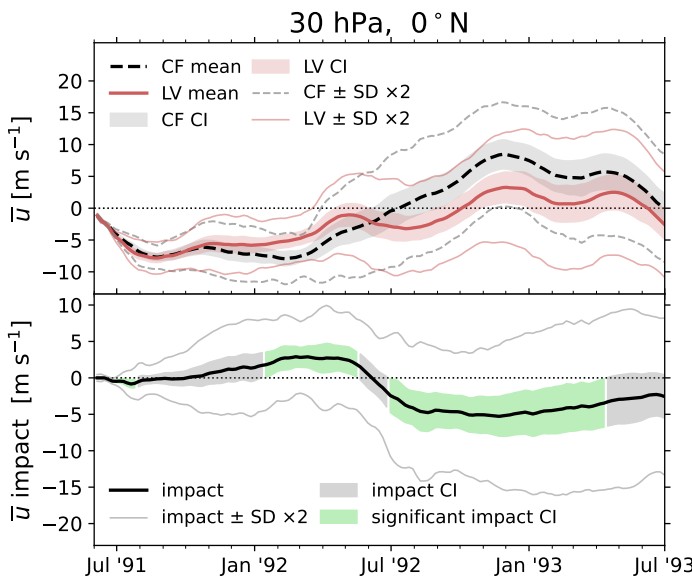

**Figure 10.** Time series of the CF ensemble-mean $\overline{u}$ and LV ensemble-mean $\overline{u}$ (upper panel) as well as the ensemble-mean $\Delta\overline{u}$ (lower panel) at 30 hPa, averaged over 5°S–5°N. In both panels, shaded bands give the confidence intervals, as computed in the way of Sect. 3.1, and thin lines give two standard deviations on each side of the mean. In the lower panel, the confidence interval is shaded in light green where the impact is statistically significant.

the SAO. For the remainder of this study, we choose to ignore the SAO impacts for brevity and instead focus on the response of the QBO.

In our simulations, the Pinatubo eruption occurs during a descending easterly QBO phase, with easterly wind shear (see Fig. 2). This state persists for around 6 months, at which point a westerly shear develops in January 1992, and westerly winds descend to 30 hPa by July 1992. Figure 4(e) shows that a significant positive $\Delta\overline{u}$ occurs in the tropics from December 1992 to May 1992, and subsequently a significant negative $\Delta\overline{u}$ occurs from July 1992 to April 1993. We will refer to these features as the "westerly response" and the "easterly response", respectively. Both the westerly and easterly responses oppose their simultaneous QBO phase. Fig. 4 panel (b) suggests that the westerly response is a part of a broader summer impact in the low latitudes of the SH, while panel (c) shows that the easterly response is a localized feature which alters the QBO in particular. Together, we interpret these impacts as a significant (yet subtle) weakening of the QBO during the 2-year post-eruption period.

This QBO weakening is seen clearly in Fig. 10, which shows the separation of the LV and CF ensembles in one-dimension as $\overline{u}$ and $\Delta\overline{u}$ at 30 hPa, averaged over 5°S–5°N. The standard deviation and confidence intervals are analogous to those shown in Fig. 5. This indicates that the easterly and westerly QBO phases are weakened by $\sim$3 m s$^{-1}$ and $\sim$5 m s$^{-1}$, respectively (which approaches 50% of the CF ensemble mean values). The arrival of the westerly phase at 30 hPa also appears delayed by $\sim$2-3 months.





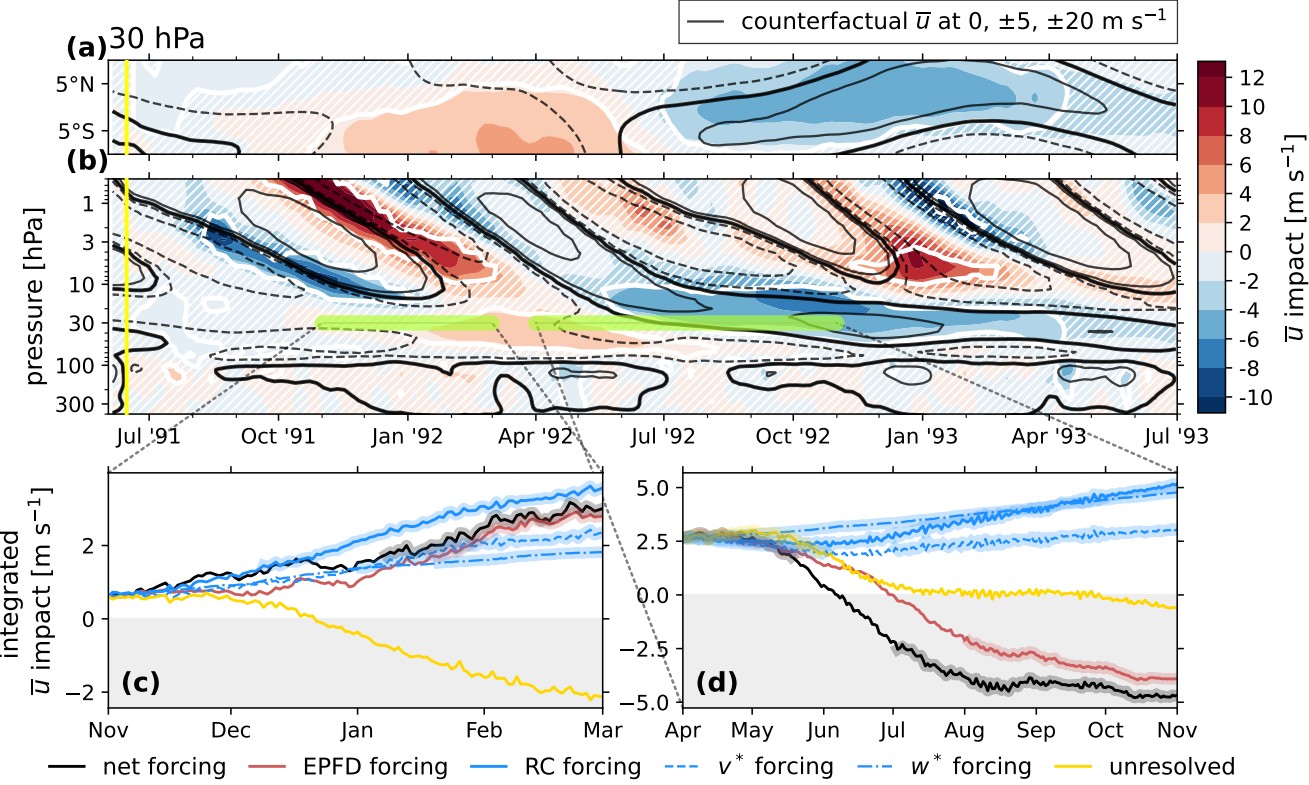

**Figure 11.** The same as Fig. 6, with the following modifications: panels (a), (c), and (d) are shown for the 30 hPa level. In panels (b), (c), and (d), the latitude band average is taken over 5°S–5°N. The CF $\overline{u}$ contours in panels (a) and (b) are drawn at 0, ±5, and ±20 m s$^{-1}$.

### 5.2.1 TEM Balance of the Tropics

We now investigate the impacts to the TEM balance associated with the weakened QBO. Figure 11 picks out the 5°S–5°N

latitude band from Figure 4(e), and reproduces it in its panel (a). Panel (b) shows the vertically-resolved zonal-wind impacts averaged over this band. In the same way as Fig. 6, the integrated impacts of the individual TEM forcing terms are shown as functions of time over two targeted time windows. The first window (panel (c)) is positioned to study the TEM imbalance which generates the westerly $\Delta\overline{u}$ during the easterly QBO phase. The second window (panel (d)) is positioned to show the TEM imbalance which drives the transition from a westerly to easterly $\Delta\overline{u}$, as the westerly QBO phase descends.

Figure 11(c) indicates that the westerly response is driven by some combination of decreased large-scale wave drag (a positive EPFD impact) and an increased forcing by the residual circulation, though this effect is opposed by unresolved sources, implying enhanced gravity wave drag. Because of the high-Rossby-number environment in the tropics, forcing by the residual circulation is primarily advective. There are approximately equal contributions from $v^*$ and $w^*$, as the zero-line of the streamfunction $\Psi^*$ is generally located far from the equator at 30 hPa except near the equinoxes (see Fig. 1(a)–(d)). On the other





hand, Fig. 11(d) shows that the transition from the westerly response to the easterly response, and thus the opposition to the descent of the westerly QBO phase, is due primarily to enhanced resolved and unresolved wave drag at 30 hPa, and has only an mild inhibiting contribution from residual circulation advection.

For a more complete picture of these mechanisms of impact, we performed an analysis of the complete TEM budget and its associated impacts in the meridional plane, averaged over one-month periods, exactly analogous to that presented in Sect. 5.1.1.

The averaging periods were again chosen to align with times of most rapid change in Fig. 11(c, d) (i.e. strongest TEM tendency impacts). These are shown in Fig. S1 and Fig. S2, provided in a supplement to this article. The main finding was further evidence that an increase in large-scale and gravity wave drag drives the transition from the westerly to easterly response, in agreement with Fig. 11. We did not pursue this further, and leave the task of examining the volcanic modification of large-scale tropical wave activity to future work; a subject which has received much less attention in the literature than the comparable midlatitude

problem.

## 6   Discussion of Comparable Results in the Literature

As noted in Sect. 5.1.1, the winter-time wave-deflection mechanism that we have identified is consistent with that proposed by Bittner et al. (2016). In that study, the effect is showcased for a much larger 55 Tg eruption. The authors noted, but did not show, that the effect was also detectable in their Pinatubo-sized eruption simulations, but was less robust. The fact that we

are able to robustly identify the wave-deflection for a 10 Tg eruption may be due to several factors. First, the model employed by Bittner et al. (2016) was described as showing excessive interannual variability compared to observations, which as they suggest, "masks the response to the forcing of the volcanic aerosols". To demonstrate this, they show that while a 55 Tg eruption was able to force both a positive anomaly in the mean *and* a negative anomaly in the variability of the vortex region, a Pinatubo-sized eruption was only able to force the mean. In fact, they show that variability of the vortex instead *increased*

during the winter of 1992, suggesting that the vortex had not been coherently forced across the ensemble members, despite the departure of the ensemble mean.

In our experiments, we don't observe the same masking by internal variability, at least not to the same extent, nor do we draw the same conclusion about the variability anomaly in the vortex region. Figure 12 compares the northern-hemisphere variability between the CF and LV ensemble members. The top panel shows a time series of the standard deviation in $\overline{u}$ at 10 hPa, averaged

over 30–50°N (matching the latitude band of Fig. 7). This shows that the midlatitude variability in the LV ensemble does exhibit intermittent increases during the winter of 1992. However, in the bottom panel, we see that the largest increases in variability are located at the lower poleward edge of the vortex (near 75°N, 3–30 hPa), and are not necessarily associated with the significant westerly response that we observed. Within the region of significant $\Delta\overline{u}$ (red contours), the variability either increases by less than 1 m s$^{-1}$, or decreases. The variability also decreases near the vortex core (near 35°N, 0.3–3 hPa). Indeed, comparing our

Fig. 7(a) and Fig. 12 to Fig. 4 and Fig. 10 in Bittner et al. (2016) suggests that our more robust detection of the wave-deflection mechanism is owed to a relative lack of interference from background internal variability.



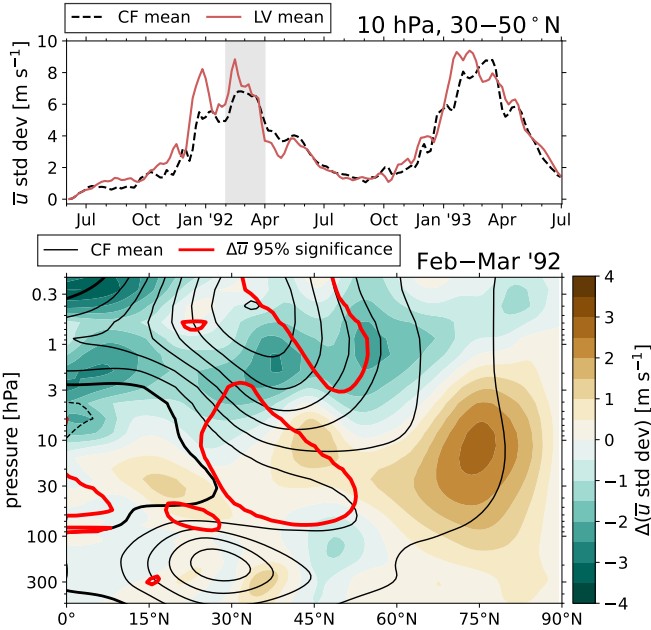

**Figure 12.** Variability of the polar vortex region in the LV and CF ensembles. **(top)** standard deviation of $\overline{u}$ at 10 hPa, averaged from $30°$ to $50°$N, for the LV (red solid) and CF (black dashed) ensemble means. **(bottom)** the impact (difference between LV and CF ensemble means) of the $\overline{u}$ standard deviation averaged over February and March 1992, as filled contours (colorscale). The time average was chosen to match Fig.7(a), and is indicated by a grey band in the top panel. Also plotted is the CF ensemble mean $\overline{u}$ (black contours), with negative values dashed and the zero-line in bold. Solid red contours show the $\Delta\overline{u}$ statistical significance (i.e. the bold white contours of Fig.7(a)) for reference.

Bittner et al. (2016) state that the reason for the increases of vortex-region variability in their volcanic simulations is unclear. At present, we can only speculate as to the relevant dynamical differences between their model and E3SMv2-SPA. It may be that the QBO is playing a role. Because an easterly QBO phase exists throughout the winter of 1992 in our simulations, we
might expect that the vortex should be weakened, and thus destabilized, by the Holton-Tan effect (Holton and Tan, 1980; Lu et al., 2020). However, Fasullo et al. (2024) recently demonstrated that in general, E3SMv2 fails to represent any Holton-Tan-type coupling between the QBO and the extratropics—that is, winter vortex speeds do not decrease relative to a climatological average during easterly QBO states. Not only does this suggest lesser vortex variability than expected, but it also hints that the background wave activity needs to be fully understood if we are to properly interpret significant post-eruption EP flux
anomalies. As the Holton-Tan effect essentially describes QBO-induced changes in Rossby waveguiding in the midlatitudes, it's plausible that the strength (or lack thereof) of the effect could directly interfere with the volcanic aerosol wave-deflection mechanism. A potential avenue for future works is to investigate this interaction more directly, by inspecting e.g. post-eruption anomalies in potential vorticity and Rossby refraction index distributions. Similar work has recently been done in a stratospheric aerosol injection (SAI) geoengineering context (Karami et al., 2023).





On another note, Post-Pinatubo summer-time anomalies have received less attention in the literature. In their idealized model studies, DallaSanta et al. (2019) observed that the quiescent summer stratosphere experiences a westerly forcing similar to that of the winter vortex region, which they attribute generically to eddy-driven effects. Toohey et al. (2014) gave more specific consideration to westerly anomalies in the austral summer of 1991 through their simulations with prescribed volcanic forcing. They observed that these anomalies were associated with a statistically significant increase in large-scale wave drag (EPFD),

and an accelerated residual circulation (consistent with our Fig. 6), though they did not show how these effects contribute to the TEM momentum budget. Moreover, Bittner et al. (2016) did not analyze summer-time anomalies in either hemisphere, and so it has been left unclear whether or not some form of wave-deflection effect acts there as well. Our study has shown that if a similar wave-driving is present, it does not rise to a level that exerts control over the manifest wind anomalies. Rather, the summer-time response is primarily driven by enhanced Coriolis torque due to an acceleration of the advective BDC.

In the tropics, our results read differently than the studies of Brown et al. (2023), though they are not necessarily contradictory. In that paper, the authors describe that the fundamental interaction between volcanic aerosol forcing and the QBO is an increased tropical upwelling, which slows the descent of the coming QBO phase. Our results agree that enhanced diabatic upwelling is the dominant initial response in the tropical region (Fig. 8(c)), but this does not immediately result in robust impacts on the QBO itself. This is likely because the Pinatubo eruption is offset from the equator, near $15°$N, whereas the

aerosol injection used by Brown et al. (2023) was on the equator. However, by the time significant tropical impacts do arise in our simulations (during boreal spring of 1992) the TEM balance does not indicate that anomalous diabatic vertical motion is responsible. In fact, the weakening of the QBO amplitude and delay of the westerly phase descent during boreal summer of 1992 is due largely to increased large-scale and gravity wave drag (Fig. 11).

    This apparent contradiction is resolved by the differences in the analysis framework. It must be emphasized that what

Brown et al. (2023) calls "vertical momentum advection" is specifically $w \times \partial u/\partial z$. This is not the same as $w^* \times \partial u/\partial z$; the difference is a contribution of the meridional derivative of the eddy streamfunction $\psi$ (Eq. (10)). In other words, an Eulerian-mean upwelling impact $\Delta w$ evaluated on a pressure-based vertical coordinate will always be some combination of eddy effects and residual-circulation advection from a TEM perspective. Our results indicate that the former can often dominate over the latter in controlling the response of the QBO region. Given the extent of our analysis, we are unable to determine if this

conclusion is generic, or if it is particular to an easterly-to-westerly QBO transition. It also may be that the conclusion would change for anomalies that occur closer in time to the eruption.

## 7    Conclusions

The goal of this study was to use the TEM analysis framework to identify the dynamical processes in control of the development and deterioration of statistically significant zonal wind anomalies following a large tropical volcanic eruption. The

essential result is that, in E3SMv2, while midlatitude westerly anomalies appear consistently for over one year in the post-eruption atmosphere, the processes controlling those anomalies are different during the summer and winter. For Pinatubo-sized eruptions occurring during summer in the NH, the initial response is enhanced tropical upwelling, which by mass continuity





drives an accelerated BDC, and thus increased zonal wind speeds northward of 30°N (Fig. 6, Fig. 8). With these westerly anomalies established, the winter stratosphere is primed for lower-stratosphere equatorward wave-deflection, which acts to further intensify upper-level westerly winds in the vortex region (Fig. 6, Fig. 7).

The initial upwelling response is as close to a direct aerosol effect that we observed, while the subsequent midlatitude accelerations are indirect consequences of the modified mean meridional circulation, and wave propagation. Moreover, the initial upwelling effect does not result in robust wind responses even in the tropical region. Rather, we did not identify significant volcanic impacts on the QBO until the winter of 1992, which were largely wave-driven effects in opposition of the simultaneous QBO phase (Fig. 11).

Our specific conclusions are as follows:

1. We concur with Bittner et al. (2016) that large-scale equatorward wave deflection near the surf zone contributes to post-eruption westerly anomalies in the winter vortex region.

2. We found that the wave-deflection argument holds only in the winter hemisphere. Westerly anomalies in the midlatitude summer stratosphere are instead explained almost entirely by an accelerated residual circulation, and thus an anomalous Coriolis forcing of momentum.

3. We observed that a weakened QBO amplitude and a delayed descent of the westerly QBO phase during summer of 1992 was caused primarily by enhanced large-scale and gravity wave drag near 30 hPa. This finding is not necessarily inconsistent with previous studies that instead emphasize the role of perturbed tropical upwelling, as noted in Sect. 6. Effectively, we have identified that the anomalous Eulerian-mean upwelling of momentum affecting the QBO in this context is eddy-driven, rather than simply an enhanced diabatic vertical motion.

4. Practically, we find that regions of robust zonal-wind anomalies can indeed be associated with equally robust forcing anomalies as diagnosed in a TEM framework. However, co-location of these wind and forcing anomalies in space and time is not required; regions where forcing anomalies are insignificant can give rise to significant wind responses, and those responses, once established, may persist after the forcing anomaly ceases.

To our knowledge, this is the only work after Bittner et al. (2016) to demonstrate the wave-deflection pathway in the post-eruption winter stratosphere, and the first to provide a full-accounting of the TEM momentum balance during specific periods of anomalous stratospheric zonal-winds in a volcanic forcing experiment. At this point, we see at least two directions for continued research. As we did not know whether robust wave-driven mechanisms of impact would arise in our simulations, our analysis stopped short of investigating the origin and behavior of the anomalous wave activity in more detail, which could be pursued (see discussion in Sect. 6). In an upcoming work using these same datasets, we will explore the volcanic impacts to the residual streamfunction more directly, as well as the resulting pertubations to stratospheric transit times of trace gases, and stratosphere-troposphere mass exchange.





*Code and data availability.* Data from the full E3SMv2-SPA simulation campaign including pre-industrial control, historical, and Mt.
Pinatubo ensembles will be hosted at Sandia National Laboratories with location and download instructions announced on
https://www.sandia.gov/cldera/e3sm-simulations-data/ when available. Code for computing the TEM quantities (Sect. 3.2), as well as the
spherical harmonic zonal averaging routine (Sect. B) are available at a public repository at https://github.com/jhollowed/PyTEMDiags. A
frozen version of the repository as of the submission of this manuscript is alternatively available at https://zenodo.org/records/15190910.

## Appendix A:  Numerical Recipes for Tendencies and Their Integration

The total tendency in the zonal-mean zonal wind $\overline{u}_i$ at time $t_i$ is computed by the first-order accurate forward finite difference

$$\frac{\partial \overline{u}_i}{\partial t} \approx \frac{\overline{u}_{i+1} - \overline{u}_i}{\Delta t} \tag{A1}$$

for an integer $i \in [1, N]$, where $N$ is the total number of time samples in the dataset, $\Delta t = (t_{i+1} - t_i)$, and $\overline{u}_i \equiv \overline{u}(t_i)$. Occurrences of $\Delta$ in this appendix take the conventional meaning, rather a notation of impact as in the rest of this paper.

For the analysis presented in Sect. 5, we compute the integrated tendency of a particular component $x$ of the TEM momentum
budget (those terms on the right-hand side of Eq. (6)) as

$$\overline{u}_i^{(x)} \equiv \overline{u}_n + \Delta t \sum_{i'=n}^{i} \frac{\partial \overline{u}_{i'}}{\partial t} \bigg|_{(x)} \tag{A2}$$

for an integer $i \in [n, N]$. This form estimates the total "accumulated" eastward wind speed by the forcing mechanism $x$ over the time period from $t_n$ to $t_i$, given a common initial condition $\overline{u}_n$. If this recipe is applied to the *total* tendency $\partial \overline{u}/\partial t$, then Eq. (A2) is just a reversal of the finite difference procedure Eq. (A1), and the model data $\overline{u}_i$ is recovered exactly. That is,
starting the integration from a certain time $n$ does not alter the time series $\overline{u}$, as expected.

For the TEM components, on the other hand, the choice of $n$ does make a material difference to the time series $\overline{u}^{(x)}$. This is a important step in the analysis, since there is strong cancellation between the residual circulation and the EP-flux divergence forcings, especially in the midlatitudes. There, the net resulting zonal-wind is only a relatively small imbalance between these effects. In addition, there is a seasonal asymmetry in the both the residual circulation speed and large-scale wave activity, such
that more zonal wind speed is accumulated by each of these forcing sources during the winter than the summer. Thus, applying Eq. (A2) with $n = 1$ results in two time series $(\overline{u}^{(v^*)} + \overline{u}^{(w^*)})$ and $\overline{u}^{(\nabla \cdot \mathbf{F})}$ which simply diverge across the dataset. Choosing $n > 1$ instead calibrates the data to a time period of interest.

In order to apply the significance tests of Sect. 3.1 to the resulting integrated tendencies, Eq. (A2) is computed per-ensemble-member, for all $x$ and for each unique choice of $n$. Note that because the unresolved forcing term $\overline{X}$ (Eq. (18)) closes the
momentum budget by construction, the sum of $\overline{u}_i^{(x)}$ over all of the TEM components $x$ also recovers the model data $\overline{u}_i$ exactly.





For the meridional and vertical gradients required for the TEM equations of Sect. 3.2, we use second-order accurate central differences for interior points, and either forward or backward finite differences at the boundaries. For a quantity $f(\phi_i)$, this is

$$\frac{\partial f_i}{\partial \phi} \approx \begin{cases} \frac{f_{i+1}-f_i}{\phi_{i+1}-\phi_i} & i=1 \\ \frac{f_{i+1}-f_{i-1}}{\phi_{i+1}-\phi_{i-1}} & 1<i<N \\ \frac{f_i-f_{i-1}}{\phi_i-\phi_{i-1}} & i=N. \end{cases} \tag{A3}$$

An analogous form is used for $\partial f/\partial p$. Note that the TEM equations only require meridional derivatives of zonally-averaged

quantities. Therefore, the recipe Eq. (A3) need only be applied to the data after remapping to a uniform latitude grid via Eq. (B9), and is not used on the native grid.

## Appendix B: Spherical Harmonic Zonal Averaging

For taking zonal averages, rather than interpolating the data to a latitude-longitude grid, we use a spectral method which allows us to obtain atmospheric variable eddy components on the native cubed-sphere simulation grid. For a 2D scalar $A(\phi, \lambda)$, we

may write it's zonal average $\overline{A}$ as a superposition of orthonormal basis functions:

$$\overline{A}(\phi) = \sum_{l=0}^{\infty} b_l Y_l^0(\phi) \tag{B1}$$

Here, $b_l$ are coefficients which scale each contributing basis function, and the basis functions are the set of zonally-symmetric spherical harmonics $Y_l^{m=0}$ of degree $l$:

$$Y_l^0(\phi) = \sqrt{\frac{2l+1}{4\pi}} P_l(\cos\phi). \tag{B2}$$

where $P_l(\phi)$ are the Legendre polynomials. To justify this definition of $\overline{A}$, it can be shown that decomposing the dataset $A(\phi, \lambda)$ into the full set of spherical harmonics $Y_l^m(\phi, \lambda)$ for $0 \leq l < \infty$ and $-l \leq m \leq l$, and zonally-averaging the resulting superposition causes all of the non-symmetric terms ($m \neq 0$) to vanish, yielding Eq. (B1).

Computationally, this expression for $\overline{A}$ can be approximated for a finite set of basis functions on $0 \leq l \leq L$ as

$$\overline{A}(\phi) \approx \sum_{l=0}^{L} b_l Y_l^0(\phi) \tag{B3}$$

Consider a sampling of $A$ resulting in the data vector $\mathbf{A}$ defined on a discrete set of $N$ gridpoints with latitudes $\phi_i$. The zonal-mean of this data, $\overline{\mathbf{A}}$ is found by solving for the coefficients $b_l$ at each $\phi_i$. The spherical harmonic amplitudes for each $(\phi, l)$ pair are stored in an $N \times (L+1)$ matrix $\mathbf{Y}_0$, and the coefficients $b_l$ in a column vector $\mathbf{B}$:

$$\mathbf{Y}_0 = \begin{array}{c} \\ \phi_1 \\ \phi_2 \\ \vdots \\ \phi_N \end{array} \begin{array}{cccc} l=0 & l=1 & \dots & l=L \\ \left[ \begin{array}{cccc} Y_0^0(\phi_1) & Y_1^0(\phi_1) & \dots & Y_L^0(\phi_1) \\ Y_0^0(\phi_2) & Y_1^0(\phi_2) & \dots & Y_L^0(\phi_2) \\ \vdots & \vdots & \ddots & \vdots \\ Y_0^0(\phi_N) & Y_1^0(\phi_N) & \dots & Y_L^0(\phi_N) \end{array} \right] \end{array}, \quad \mathbf{B} = \begin{bmatrix} b_0 \\ b_1 \\ \vdots \\ b_L \end{bmatrix} \tag{B4}$$





In matrix notation, Eq. (B3) is

$$\overline{\mathbf{A}} \approx \mathbf{Y}_0 \mathbf{B}, \tag{B5}$$

and the coefficients $\mathbf{B}$ are solved for by decomposition of the native grid data $\mathbf{A}$ onto the zonally-symmetric basis set. This requires inversion of $\mathbf{Y}_0$:

$$\mathbf{B} = \mathbf{Y}_0^{-1} \mathbf{A} \tag{B6}$$

and so

$$\overline{\mathbf{A}} \approx \mathbf{Y}_0 \mathbf{Y}_0^{-1} \mathbf{A} \tag{B7}$$

The inversion $\mathbf{Y}_0^{-1}$ can be obtained by computing a least-squares solution to the equation $\mathbf{Y}_0^{-1} \mathbf{Y}_0 = \mathbf{I}$, where $\mathbf{I}$ is the $(L \times L)$ identity matrix. Alternatively, it can be obtained by $\mathbf{Y}_0^{-1} \equiv \mathbf{Y}_0^{\mathrm{T}} \mathrm{diag}(\mathbf{w})$ where $\mathrm{diag}(\mathbf{w})$ is an $(N \times N)$ matrix with a vector $\mathbf{w}$ of $N$ data weights on the diagonal. The weights provide the areas of the model grid cells, normalized such the the sum of the weights is $4\pi$.

This procedure results in zonal-means, and thus eddy components, at all grid points $N$. Alternatively, an analogous procedure can be followed to recover the zonal mean $\overline{\mathbf{A}}$ on a different, coarser latitude grid. This is more appropriate for visualization and storage of the zonal-mean components. Given a uniform set of latitudes $\phi'$ of length $M \ll N$, we store the $l = 0$ spherical harmonics at those latitudes in an $M \times (L+1)$ matrix $\mathbf{Y}_0'$:

$$\mathbf{Y}_0' = \begin{array}{c} \\ \phi_1' \\ \phi_2' \\ \vdots \\ \phi_M' \end{array} \begin{array}{cccc} l=0 & l=1 & \dots & l=L \\ \left[ \begin{array}{cccc} Y_0^0(\phi_1') & Y_1^0(\phi_1') & \dots & Y_L^0(\phi_1') \\ Y_0^0(\phi_2') & Y_1^0(\phi_2') & \dots & Y_L^0(\phi_2') \\ \vdots & \vdots & \ddots & \vdots \\ Y_0^0(\phi_M') & Y_1^0(\phi_M') & \dots & Y_L^0(\phi_M') \end{array} \right] \end{array} \tag{B8}$$

and the zonal mean of the data is again a composite of these basis functions,

$$\overline{\mathbf{A}}' \approx \mathbf{Y}_0' \mathbf{B} = \mathbf{Y}_0' \mathbf{Y}_0^{-1} \mathbf{A}. \tag{B9}$$

This equation transforms the data from a $N$-point grid in real space, to the set of $L+1$ coefficients $b_l$ in spectral space, before transforming back to a $M$-point grid in real space.

The matrices $\mathbf{Y}_0$, $\mathbf{Y}_0^{-1}$, and $\mathbf{Y}_0'$ need only be computed once for a given pairing of computational grid $\phi$, remap grid $\phi'$, and choice of $L$, after which they can be saved to file for use in subsequent computations.

## Appendix C: Physically Consistent Vector-Field Representation of the Meridional Circulation

It is common in the literature for authors to visually represent the circulation in the meridional-vertical plane as a vector field with components $(\overline{v}, \overline{w})$. However, because meridional velocities are typically several orders of larger than vertical velocities, $\overline{w}$





is often multiplied by a post-hoc scaling factor. This scaling factor is usually of order 100, though the specific choice depends
on the data localization and processing. In any case, the choice is usually not explicitly justified, and it is often difficult to
discern if the direction of the vectors are providing the correct physical picture of the circulation.

  Inspired by the scaling recommendations of Jucker (2021) for the EP-flux vectors, here we introduce a method of con-
structing a vector field of the meridional circulation which is physically-consistent with rotation in the meridional-vertical
plane. Specifically, we define a vector field which is everywhere tangent to isolines in a mass streamfunction $\Psi$ (in the form of
Eq. (12), with a sign convention such that positive structures in $\Psi$ indicate clockwise rotation, and vice-versa. In other words,
the curl of the visualized vector field will "look" correct, and regions of zero curl with correspond to $\Psi = 0$. This is usually the
implicit goal of manual scaling of the vertical velocity $w$.

  We will assume that the data is provided in a $\phi$-$p$ plane, with latitude $\phi$ in degrees and pressure $p$ in hPa, with $p$ decreasing
toward the positive end of the vertical axis. The gradient of $\Psi$ in data units is

$$
\quad \nabla\Psi = \begin{bmatrix} \partial\Psi/\partial\phi \\ \partial\Psi/\partial p \end{bmatrix}, \tag{C1}
$$

which we compute by the finite difference methods of Appendix A. With $\Psi$ given in kg s$^{-1}$, the units of the meridional and
vertical components of $\nabla\Psi$ are kg s$^{-1}$ deg$^{-1}$ and kg s$^{-1}$hPa$^{-1}$, respectively. Because these derivatives are in different units,
they must be scaled such that plotting the vector field $\nabla\Psi$ shows the correct direction and amplitude at any location in the plot.

  This amounts to a coordinate transformation from data units to "display units". Following Jucker (2021), we define the
display coordinates in the plot as $X$ and $Y$, respectively, which have units of length (e.g. inches). We also define a pair of
normalized coordinates $x, y \in [0, 1]$ which give the fractional distance along each axis. For any other coordinate $\alpha \in [\phi, X]$
mapping to $d \equiv x$, or $\alpha \in [p, Y]$ mapping to $d \equiv y$, we use the notation $\alpha_0 \equiv \alpha(d = 0)$, $\alpha_1 \equiv \alpha(d = 1)$, and $\Delta\alpha \equiv \alpha_1 - \alpha_0$
(occurrences of $\Delta$ in this appendix take the conventional meaning, rather a notation of impact as in the rest of this paper). If
the axes are linear, then the data and display units are related by

$$
\quad \phi(X) = \frac{\Delta\phi}{\Delta X}X + \phi_0 \tag{C2}
$$

$$
p(Y) = \frac{\Delta p}{\Delta Y}Y + p_0 \tag{C3}
$$

The streamfunction gradient in the display units is then obtained via the transformations

$$
\frac{\partial\Psi}{\partial X} = \frac{\partial\Psi}{\partial\phi}\frac{\partial\phi}{\partial X} = \frac{\partial\Psi}{\partial\phi}\frac{\Delta\phi}{\Delta X}, \tag{C4}
$$

$$
\frac{\partial\Psi}{\partial Y} = \frac{\partial\Psi}{\partial p}\frac{\partial p}{\partial Y} = \frac{\partial\Psi}{\partial p}\frac{\Delta p}{\Delta Y}, \tag{C5}
$$

Since we can freely make scaling changes to the components which maintain the vector lengths to a constant factor, we will
express this as

$$
\nabla\Psi_{(X,Y)} \equiv \begin{bmatrix} \Delta X \times \partial\Psi/\partial X \\ \Delta X \times \partial\Psi/\partial Y \end{bmatrix} = \begin{bmatrix} \Delta\phi\,\partial\Psi/\partial\phi \\ \Delta p(\Delta X/\Delta Y)(\partial\Psi/\partial p) \end{bmatrix} \tag{C6}
$$



where $\Delta X/\Delta Y$ is the plot aspect ratio. Finally, we obtain the tangent vector field $\nabla\Psi^{\perp}$ by swapping the components and applying the desired sign convention:

$$\nabla\Psi^{\perp}_{(X,Y)} = \begin{bmatrix} -\Delta p(\Delta X/\Delta Y)(\partial\Psi/\partial p) \\ \Delta\phi\,\partial\Psi/\partial\phi \end{bmatrix} \tag{C7}$$

This is the vector field that we plot for visualizing the meridional circulation on linear latitude-pressure axes. Note that if the display coordinates $X$ and $Y$ are expressed in e.g. inches, then Eq. (C4) and Eq. (C5) have units of kg s$^{-1}$ in$^{-1}$. The components of Eq. (C6) and Eq. (C7) thus have units of kg s$^{-1}$ y-unit$^{-1}$, where y-unit represents a unit-length of the y-axis, defined by $\Delta Y = 1$ y-unit.

If we are instead visualizing the vector field in a log-pressure vertical coordinate $z = \log_{10}(p)$, then a different transformation for the vertical component is required. If we define a normalized coordinate $y \in [0,1]$, giving the fractional length along the vertical axis, then the relation between $y$ and $p$ is given by Jucker (2021) as

$$y(p) = \frac{\log_{10}(p_0) - \log_{10}(p)}{\log_{10}(p_0) - \log_{10}(p_1)} = \frac{\ln(p_0/p)}{\ln(p_0/p_1)} \tag{C8}$$

which implies

$$p(y) = p_0\,(p_1/p_0)^y. \tag{C9}$$

The derivative of $p$ with respect to the display coordinate $Y$ is then

$$\frac{\partial p}{\partial Y} = \frac{\partial p}{\partial y}\frac{\partial y}{\partial Y} = \frac{1}{\Delta Y}\frac{\partial p}{\partial y} \tag{C10}$$

and Eq. (C5) is replaced with

$$\frac{\partial\Psi}{\partial Y} = \frac{\partial\Psi}{\partial p}\frac{\partial p}{\partial Y} = \frac{\partial\Psi}{\partial p}\frac{1}{\Delta Y}\left(p_0\ln\left(\frac{p_1}{p_0}\right)\left(\frac{p_1}{p_0}\right)^y\right). \tag{C11}$$

where $y$ in this equation is the evaluation of Eq. (C8) at $p$. Following the procedure in Eq. (C6)–Eq. (C7) above, the vector field that we plot for visualizing the meridional circulation on latitude-log-pressure aces is thus

$$\nabla\Psi^{\perp}_{(X,Y)} = \begin{bmatrix} -\frac{\Delta X}{\Delta Y}\frac{\partial\Psi}{\partial p}\left(p_0\ln\left(\frac{p_1}{p_0}\right)\left(\frac{p_1}{p_0}\right)^y\right) \\ \Delta\phi\,\partial\Psi/\partial\phi \end{bmatrix} \tag{C12}$$

Figure C1 shows the effect of these different choices for plotting the gradient-normal vector field of $\Psi$ for a logarithmic vertical pressure axis, with and without the plot aspect ratio scaling and derived log-pressure scaling. This demonstrates that for rendering a vector field parallel to isolines of $\Psi$ while maintaining physically correct vector directions in plots of arbitrary shape, the scalings presented here are the correct ones.

*Author contributions.* JH wrote the analysis codes, processed and analyzed the ensemble data, created figures and wrote the manuscript. CJ carefully advised the research, and assisted with the implementation and understanding of the TEM analysis framework. TE, BW, and DB





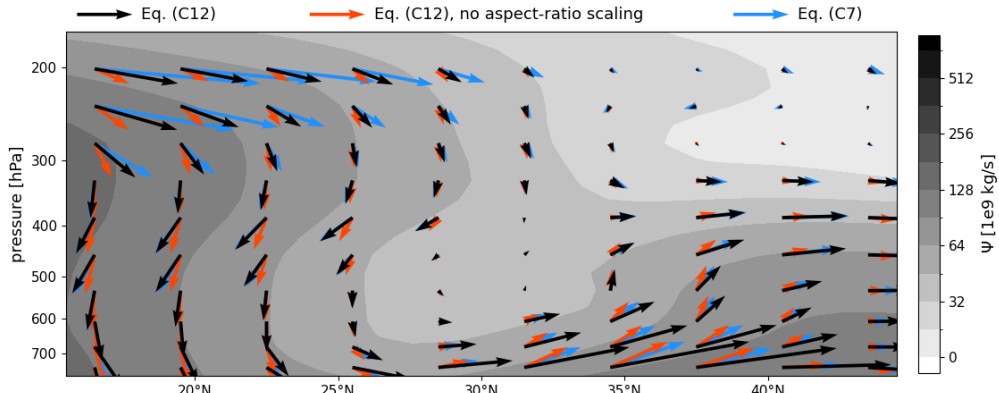

**Figure C1.** A comparison of three representations of the gradient-normal vector field of a winter-time streamfunction $\Psi$ in the northern hemisphere on a log-scaled vertical pressure axis, with a figure aspect ratio of 2.4. $\Psi$ is shown in the greyscale contours. Black arrows show the vectors tangent to isolines in $\Psi$, as computed by Eq. (C12). Red arrows show the same vectors, but with the aspect-ratio scaling removed (the horizontal component of Eq. (C12) is multiplied by $\Delta Y / \Delta X$). Blue arrows shows the vectors as computed by Eq. (C7) without the log-pressure correction. The black arrows are the only ones which are everywhere tangent to the $\Psi$ contours. The blue arrows would be tangent if the vertical pressure axis were linearly-scaled. The vector lengths correspond to the magnitude $\nabla \Psi$ at each point. The arrows presented in the figure legend represent $4 \times 10^{11}$ kg s$^{-1}$ per unit length of the y-axis.

designed and ran the E3SM simulation ensembles, and provided invaluable assistance on the data usage and analysis. TE in particular was
helpful in testing and interpreting the TEM analysis results, and was the original author of the spherical-harmonic-based zonal averaging scheme of Appendix B. DB provided frequent and thoughtful feedback on the research, and contributed text to the manuscript. BH provided simulation and software support, and participated in the study design. All authors read and edited the manuscript.

*Competing interests.* The authors declare that they have no conflict of interest

*Acknowledgements.* This research benefited from discussions with the CLDERA team at SNL, with software and computing support in
particular from Jerry Watkins and Kara Peterson, as well as helpful discussions with UM scientists Owen Hughs and Aaron Johnson. The work was supported by the Laboratory Directed Research and Development program at Sandia National Laboratories (SNL), a multimission laboratory managed and operated by National Technology & Engineering Solutions of Sandia, LLC, a wholly owned subsidiary of Honeywell International Inc., for the U.S. Department of Energy's National Nuclear Security Administration under contract DE-NA0003525. This written work is co-authored by employees of NTESS. The employees, not NTESS, owns the right, title and interest in and to the written work
and is responsible for its contents. Any subjective views or opinions that might be expressed in the written work do not necessarily represent the views of the U.S. Government. The publisher acknowledges that the U.S. Government retains a non-exclusive, paid-up, irrevocable,





world-wide license to publish or reproduce the published form of this written work or allow others to do so, for U.S. Government purposes. The DOE will provide public access to results of federally sponsored research in accordance with the DOE Public Access Plan. The University of Michigan (UM) researchers were supported by an SNL subcontract, award number 2305233. This research used resources of the National

Energy Research Scientific Computing Center (NERSC), a Department of Energy Office of Science User Facility using NERSC award BER-ERCAP0026535. The analyses presented made use of the NumPy (Harris et al., 2020), MetPy (May et al., 2022), and xarray (Hoyer and Hamman, 2017) Python packages.



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
