# Peer review of "Volcanic Aerosol Modification of the Stratospheric Circulation in E3SMv2 Part I: Wave-Mean Flow Interaction"

_EGUsphere, 2025_

## Author Comment (AC1)

**Authors' Response to Reviews of**

**Volcanic Aerosol Modification of the Stratospheric Circulation in E3SMv2 Part I: Wave-Mean Flow Interaction**

Joseph Hollowed, Christiane Jablonowski, Thomas Ehrmann, Diana L. Bull, Benjamin Wagman and Benjamin Hillman
*egusphere-2025-1756*
* * *
RC: *Reviewer Comment*,     AR: Author Response,     ☐ Manuscript Text

**1.  Reviewer #1**

**1.1.  Author Comments**

We thank the reviewer for the careful reading of our manuscript and the useful feedback. Each comment below appears as a reviewer comment (RC) followed by an author response (AR). Closed boxes show text from the manuscript. Red text with strikethrough represents deleted text, and blue text with wavy underlining represents new text. Section numbers refer to those as they appear in the updated manuscript.

Our responses to Comment 1, 2 and 3 consist of clarifying notes to the reviewer, as well as text modifications in the manuscript per the reviewer feedback.

**1.2.  Comment 1**

RC:  *There is a limitation of only using a single model in this analysis. Are there any possible implications in your results from having a slightly weaker QBO with a phase lock?*

AR:  This is an important point. Yes, we do believe that there are possible implications of E3SMv2's QBO behavior on the results in both the tropics and midlatitudes. We attempted to discuss this as it relates to the midlatitude results in the third paragraph of Section 6. There, we noted that the lack of a Holton-Tan effect in E3SMv2 could be enhancing the significance of our findings with respect to previous studies. We did not, however, discuss this as it relates the the QBO findings specifically. The most obvious need for a comment on this matter is in the last paragraph of Sect. 6, where we say

> ...In other words, an Eulerian-mean upwelling impact $\Delta w$ evaluated on a pressure-based vertical coordinate will always be some combination of eddy effects and residual-circulation advection from a TEM perspective. Our results indicate that the former can often dominate over the latter in controlling the response of the QBO region. Given the extent of our analysis, we are unable to determine if this conclusion is generic, or if it is particular to an easterly-to-westerly QBO transition. It also may be that the conclusion would change for anomalies that occur closer in time to the eruption ...

It should be obvious (and we should have stated) that this conclusion may also change for a more realistic (stronger, slower) QBO. In particular, a change to the QBO background state may well change the manner in which the volcanic impacts manifest from a TEM perspective, if e.g. the background wave activity and/or residual velocities in the tropics change significantly with respect to their observed impacts. In that case, it is not clear if the stated result (that the impacts on eddy effects are more important that advective effects) would

be maintained or not. Since we do not have enough information to speak on those possible effects, rather than speculate we've added a brief note;

> ... In other words, an Eulerian-mean upwelling impact $\Delta w$ evaluated on a pressure-based vertical coordinate will always be some combination of eddy effects and residual-circulation advection from a TEM perspective. Our results indicate that the former can often dominate over the latter in controlling the response of the QBO region. Given the extent of our analysis, we are unable to determine if this conclusion is generic, or if it isIt may be particular to an easterly-to-westerly QBO transition, or to the kind of QBO that exists in E3SMv2 (which, as discussed in Sect. 4.2, is both weak and slow compared to nature). Since the impact significance of the TEM terms are sensitive to the strength of the background, a changing QBO amplitude could change the relative importance of each effect identified by this analysis.It also may be that the conclusion would change for anomalies that occur closer in time to the eruption. We were not able to test these questions in our current experimental setup, and there is room for future work to further clarify the interaction between the QBO and volcanic TEM anomalies in E3SMv2 or other models. It would be especially useful to perform a resampling analysis on a larger sample size to better understand the robustness of, and the importance of internal variability on the tropical anomalies ...

**1.3.  Comment 2**

**RC:** *L304: 'Most of' the declining QBO strength over the simulation can be explained by ensemble spread - so there is a part which cannot be explained?*

**AR:** This was imprecise language, and we see how it could raise a question by the reader. It would be fair to remove the words "most of" in this case. We only wrote this because we had this fact in mind; the QBO is sufficiently weak in E3SMv2 such that it's strength from period-to-period can vary notably within a single simulation (see e.g. the Yu et al. (2024) paper cited in this section, their Figure 2). However, in regards to the *coherent* decrease of of QBO amplitude that is observed in our ensemble mean over several cycles, it would be fair to attribute that feature entirely to the ensemble spread. Noting that, and noting that we already provided a citation and mentioned the deficiencies with E3SMv2's QBO in the paragraph above the one in question, we have adjusted the text as follows:

> Following the first cycle, the QBO signal in the ensemble mean weakens further, by an additional factor of $\sim$2. Most of thisThis weakening is explained by the increasing intra-ensemble spread, as the members diverge from their common initial condition ...

**1.4.  Comment 3**

**RC:** *L361 among other locations: This response looks insignificant to me. What do you interpret as less significant or more significant? (also a typo in the word 'response')*

**AR:** In this case, we should have said "insignificant" rather than "less significant". Indeed, our figures display the result of a binary significance assessment (a single contour in p-value), and thus it is not possible to read the "degree" of significance from our figures. We only meant to draw the reader's attention to the extant (but insignificant, by the plotted metric) positive impact in that region. In this specific case, we likely chose the words "less significant" since there are some very small regions of significance in the vicinity of this feature (e.g. Fig. 4(d)), though the feature as a whole does not appear robust. As the reviewer noted, there are a few other locations in the paper where we refer to a feature in the impact distribution which exists, but is not in a region of significance. While we occasionally found it useful to mention these features, the text should be

refined to clearly specify what is and is not significant. The text near L361 was changed (including the typo correction) as:

> We will refer to this impact as the "winter response" (WR), and we identify this response most closely with the claims of an "accelerated vortex region" that are well-represented in the literature, as discussed in Sect. 1. A similar (but  insignificant) response also occurs in the southern hemisphere near 50°S between June and October 1992.

**Authors' Response to Reviews of**

**Volcanic Aerosol Modification of the Stratospheric Circulation in E3SMv2 Part I: Wave-Mean Flow Interaction**

Joseph Hollowed, Christiane Jablonowski, Thomas Ehrmann, Diana L. Bull, Benjamin Wagman and Benjamin Hillman
*egusphere-2025-1756*
* * *
RC: *Reviewer Comment*,    AR: Author Response,    ☐ Manuscript Text

**1. Reviewer #2**

**1.1. Author Comments**

We thank the reviewer for the careful reading of our manuscript and the useful feedback. Each comment below appears as a reviewer comment (RC) followed by an author response (AR). Closed boxes show text from the manuscript. Red text with strikethrough represents deleted text, and blue text with wavy underlining represents new text. Section numbers refer to those as they appear in the updated manuscript.

Our responses to Comment 1, 2 and 3 consist of clarifying notes to the reviewer, as well as text modifications in the manuscript per the reviewer feedback.

**1.2. Comment 1**

RC: *Sample size. As the authors pointed out, the sample size is very important to simulate the stratospheric dynamics response to the volcanic eruption, because of the internal variability. I like the figures 5&10, where it clearly showed the statistics of two ensembles. I think a 15-ensemble member for the westerly jet response is enough, but QBO could be more complex. I recommend adding a Monte Carlo test on the sample size, say, by randomly picking up different sample size, to check if the response is still robust.*

AR: We appreciate the suggestion by the reviewer, and agree that a Monte Carlo analysis of the tropical impact uncertainty would be insightful. Certainly internal variability plays a significant role in shaping the response in this region, especially with the unstable character of the QBO in E3SMv2 as described. We have considered the idea of implementing such a test, but ultimately decided that the robustness of our results is sufficiently demonstrated through the existing ensemble spread and significance tests, given Figures 2 and 10.

At a few other locations in the paper, we suggest future work to better understand the tropical volcanic response using a TEM analysis framework, since we paid more attention to the midlatitudes. We have added a suggestion for a Monte Carlo test on a larger sample size there, which would be an important contribution. Taken together with a comment from another reviewer, the final paragraph of Sect. 6 has been updated as

> ...In other words, an Eulerian-mean upwelling impact $\Delta w$ evaluated on a pressure-based vertical coordinate will always be some combination of eddy effects and residual-circulation advection from a TEM perspective. Our results indicate that the former can often dominate over the latter in controlling the response of the QBO region. Given the extent of our analysis, we are unable to determine if this conclusion is generic,  It may be particular to an easterly-to-westerly QBO transition, or

> to the kind of QBO that exists in E3SMv2 (which, as discussed in Sect. 4.2, is both weak and slow compared to nature). Since the impact significance of the TEM terms are sensitive to the strength of the background, a changing QBO amplitude could change the relative importance of each effect identified by this analysis. It also may be that the conclusion would change for anomalies that occur closer in time to the eruption. We were not able to test these questions in our current experimental setup, and there is room for future work to further clarify the interaction between the QBO and volcanic TEM anomalies in E3SMv2 or other models. It would be especially useful to perform a resampling analysis on a larger sample size to better understand the robustness of, and the importance of internal variability on the tropical anomalies ...

**1.3. Comment 2**

**RC:** *In the recent eruption of the Hunga volcano, an equatorward shift of the jet is also observed, during austral winter. It may worth mentioning in the introduction.*

AR: We thank the reviewer for this helpful suggestion. We were aware that many studies have recently been focused on the Hunga Tonga eruption of 2022, but were not familiar with these results. After some literature review, a remark and new citations have been added to the introduction as follows:

> ... This vortex-strengthening effect is correlated to an enhanced AO (see e.g. Baldwin and Dunkerton (1999)), which itself has45 been observed by an empirical orthogonal function (EOF) analysis of sea-level pressure data following 13 major volcanic eruptions between 1873 and 2000 (Christiansen, 2008). More recently, enhanced wind speeds and equatorward shifts of the vortex have been identified for the 2022 eruption of Hunga Tonga-Hunga Ha'apai eruption (Wang et al., 2023; Yook et al., 2025)...

**1.4. Comment 3**

**RC:** *Line 115 "vertical and horizontal": to be more specific, meridional.*

AR: The text near line 115 has been changed as:

> ... The primary mechanisms controlling large-scale transport of momentum in the stratosphere are advection by the so-called residual (diabatic) circulation, as well as vertical and meridional wave propagation and dissipation. ...

---

## Author Response (AR2)

**Authors' Response to the Editor of**

**Volcanic Aerosol Modification of the Stratospheric Circulation in E3SMv2 Part I: Wave-Mean Flow Interaction**

Joseph Hollowed, Christiane Jablonowski, Thomas Ehrmann, Diana L. Bull, Benjamin Wagman and Benjamin Hillman
*egusphere-2025-1756*
* * *
**EC:** *Editor Comment*,     AR: Author Response,     □ Manuscript Text

We thank the editor for the careful reading of our manuscript and the useful feedback. This document contains our responses to comments 1, 3, 4, 6, and 8. Comments 2, 5, and 7 were minor corrections that have been implemented per the editor suggestion, and are not commented upon. Each comment below appears as a editor comment (EC) followed by an author response (AR). Closed boxes show text from the manuscript. Red text with strikethrough represents deleted text, and blue text with wavy underlining represents new text.

**Comment 1**

**EC:** *L2-4: this statement doesn't sound accurate. Even if the volcano eruptions initially happen in the tropics, much of the sulfate is transported to the midlatitude and polar regions during the gas-to-particle conversion and coagulation processes. Please state it clearly if you refer to a specific volcano during its initial eruption time.*

**AR:** We were not referring to a specific eruption when we wrote this sentence, but indeed we were referring to the initial period of a few months to a year post-eruption. This is the time period during which our statement that "the primary effect of volcanic aerosols is to heat the tropical stratosphere" is true, and also the time period over which accelerations to the winter vortex region are typically observed and discussed in the literature (and in our paper). The abstract has been tweaked to be more clear about this:

> . . . This wind response has been reproduced in some (but not all) simulated eruption studies. As the primary effect of volcanic aerosols during the initial post-eruption period is to heat the tropical stratosphere, the midlatitude zonal wind response is often explained as thermal wind effect. . . .

**Comment 3 & 4**

**EC:** *L135-136: the description of how the prognostic aerosol scheme converts SO2 gas to sulfate aerosol and how the volcanic sulfate aerosol is represented in the MAM4 scheme needs improvement. Also, the MAM4 scheme in standard E3SMv2 (Wang et al., 2020), where all aerosol components appear in the coarse mode, is different from the original MAM4 (Liu et al., 2016). Please adds a bit more clarification on how the stratospheric sulfate in represented in the E3SMv2-SPA model version.*

**EC:** *L143-144: please add a reference for the specifics of the Mt. Pinatubo emissions.*

**AR:** We thank the editor for the correction on the citation for MAM4 in E3SMv2. We have added some more precise language about the specific MAM4 implementation being used in E3SMv2-SPA, and included a reference for the SO2 emissions dataset in use. The new information added comes from Brown et al. (2024),

which describes E3SMv2-SPA. Rather than repeat all of the details which are presented in that paper, and are beyond the scope of the present work, we've also added a more explicit citation to Section 2 of Brown et al. (2024), where the stratospheric modifications made to MAM4 are described in detail.

> The numerical experiments utilized for this study were conducted in a custom version of the Energy Exascale Earth System Model version 2 (E3SMv2; Golaz et al. (2022)) called E3SMv2-SPA, described in Brown et al. (2024). While E3SMv2 describes volcanic eruptions by a prescribed forcing from the GloSSAC reanalysis dataset (Thomason et al., 2018), E3SMv2-SPA instead replaces this treatment with a stratospheric prognostic aerosol (SPA) capability, using the $SO_2$ emissions database from VolcanEESMv3.11 (a modified version of the initial release by Neely and Schmidt (2016), described in Mills et al. (2016)). Rather than prescribing stratospheric light extinction directly, SO2 is emitted as a tracer into the stratosphere, which causes the formation of sulfate aerosol according to the prognostic equations of the four-mode Modal Aerosol Module (MAM4) introduced by Liu et al. (2016) and modified for E3SMv2 by Wang et al. (2020). In E3SMv2-SPA, further modifications were made to MAM4 in order to improve the representation of stratospheric sulfate aerosols following the eruption of Mt. Pinatubo, as MAM4 was designed firstly for the fidelity of tropospheric aerosol calculations (see Brown et al. (2024), Section 2 for details). Compared to SO2 emission in standard E3SMv2, this modified configuration results in a more accurate lifetime of stratospheric sulfate aerosols following the 1991 eruption of Mt. Pinatubo, …

**Comment 6**

**EC:** *L151: Is there a justification for starting the model simulations two weeks prior to the volcanic eruption to capture the true dynamical response in the stratosphere? Please clarify it in section 2.*

**AR:** Yes, and we attempted to clarify this in the text when we said "this is enough time to allow for synoptic-scale differences between members to manifest, while the large-scale circulation remains qualitatively consistent. It is in this sense that the intra-ensemble variability is "limited"..." In other words, we manually tuned this 2-week timescale to achieve a "limited" intra-ensemble variability, as described. A shorter timescale would lead to less variability, and a longer timescale would see large-scale differences between the members. Specifically, we observed that longer timescales could result in very different transport of the initial SO2 plume. We do not describe this tuning method, since it was simply a manual "guess-and-check", which was sufficient for our purposes. We have added a few more words to make all of this a bit more clear:

> …Because the ensemble begins on June 1, 1991, the individual members have only two weeks to diverge before the eruption occurs on June 15, 1991. Through a manual tuning process on this timescale, we found that two weeks was enough time to allow for synoptic-scale differences between members to manifest, and also ensured that the large-scale circulation remains qualitatively consistent (we found that much longer timescales resulted in qualitatively different transport of the initial plume, as would be expected for independent initial conditions). It is in this sense that the intra-ensemble variability is "limited", and thus the ensemble average should capture the robust climatic response to the Mt. Pinatubo event, conditioned on the real-world initial atmospheric state. …

**Comment 8**

**EC:** *L485: It's unclear about the "several factors" mentioned here. The first factor is explicitly discussed. Where are the second, third, etc.?*

AR:  This language was not meant to imply that we have knowledge of all of the factors at play here. Rather, we said this to suggest that the one factor that is explicitly discussed may or may not be the whole story. There may be (and likely are) other differences between our model and that of Bittner et al. (20216) that we have not identified, but are playing a role. We have tweaked the language slightly to avoid the implication that we might be aware of other factors that were left undiscussed:

> The fact that we are able to robustly identify the wave-deflection for a 10 Tg eruption may be due to various factors. For example, the model employed by Bittner et al. (2016) was described as showing excessive interannual variability compared to observations...